# CAPTCHA Solving for Native GUI Agents: Automated Reasoning-Action Data Generation and Self-Corrective Training

**Yuxi Chen** [* 1] **Haoyu Zhai** [* 1] **Chenkai Wang** [* 1] **Rui Yang** [1] **Lingming Zhang** [1] **Gang Wang** [1] **Huan Zhang** [1]

## Abstract

GUI agents are rapidly shifting from multi-module pipelines to end-to-end, native vision-language models (VLMs) that perceive raw screenshots and directly interact with digital devices. Despite rapid progress on general GUI tasks, CAPTCHA solving remains a major challenge. On the other hand, although specialized CAPTCHA solving pipelines exist, they cannot handle general GUI tasks. To address this gap, we introduce ReCAP: a CAPTCHA-capable native GUI agent that solves modern, interactive CAPTCHA challenges while retaining general GUI-agent performance. We first develop a dynamic CAPTCHA system spanning seven representative CAPTCHA types, designed to stress primitive and complementary capabilities for CAPTCHA solving. Then, we develop an automated data collection and curation pipeline that generates large-scale CAPTCHA interaction trajectories paired with reasoning traces. As CAPTCHA solving often requires multi-step interaction and recovery from intermediate mistakes, we further leverage failed trajectories to construct self-correction data, training agents to reflect on errors and correct their actions online. Across synthetic and real-world test sets, ReCAP substantially improves CAPTCHA-solving success over its base agents, while maintaining strong performance on general GUI-agent benchmarks.

## 1. Introduction

CAPTCHA (Completely Automated Public Turing test to tell Computers and Humans Apart) is a critical security

---
[*]Equal contribution [1]University of Illinois Urbana-Champaign. Correspondence to: Yuxi Chen <yuxi5@illinois.edu>, Huan Zhang <huan@huan-zhang.com>. Code, model, and data are available at: https://github.com/ASTRAL-Group/ReCAP-Agent.

*Proceedings of the $43^{rd}$ International Conference on Machine Learning*, Seoul, South Korea. PMLR 306, 2026. Copyright 2026 by the author(s).

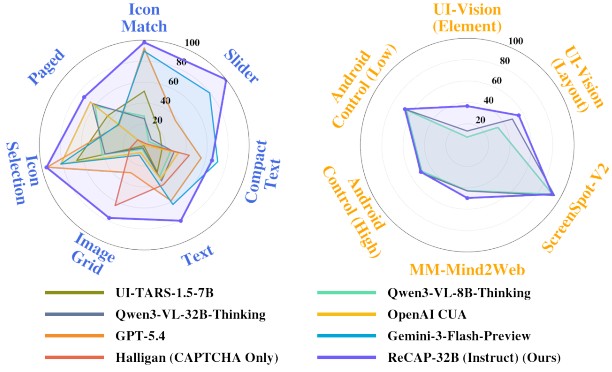

*Figure 1.* **Performance on CAPTCHA and general GUI agent benchmarks. Left:** CAPTCHA solving performance across seven challenge types in our held-out dynamic benchmark. **Right:** Performance on general GUI agent benchmarks. ReCAP-32B (Instruct) achieves the strongest overall CAPTCHA performance while maintaining comparable general GUI capabilities. We exclude GPT-5.4 and Gemini-3-Flash-Preview from the GUI evaluation as they are not fair comparison candidates for these benchmark results.

mechanism for protecting online services from automated attacks (Ahn et al., 2003). Their design has evolved rapidly in response to advances in machine learning. Modern CAPTCHAs increasingly rely on complex visual understanding and interactive behaviors like continuous control and semantic reasoning, rather than static text recognition alone (Searles et al., 2023).

Recent GUI agents including OpenCUA (Wang et al., 2025), UI-TARS (Qin et al., 2025), and Qwen3-VL (Yang et al., 2025) have demonstrated strong ability on general-purpose GUI interaction tasks, including web browsing, desktop and mobile control, and application usage. Large-scale trajectory datasets from Aguvis (Xu et al., 2024), GUI World (Chen et al., 2024), and AgentNet (Wang et al., 2025) have enabled models to acquire broad GUI grounding and interaction skills. Despite this progress, these agents are not explicitly designed for unique challenges posed by modern CAPTCHAs. Consequently, they tend to underperform on CAPTCHA solving as they lack complementary skills required for CAPTCHA solving, including robust OCR under heavy noise and various text stylization, fine-grained visual

understanding, and precise control. Solving CAPTCHA has become a challenging task for evaluating the robustness and generalization capabilities of GUI agents (Wu et al., 2025c), as CAPTCHAs may naturally appear during daily computer-use. Yet, there is a lack of public and systematic research on studying CAPTCHA solving in native GUI agents.

Several recent systems have explored general-purpose approaches to solving modern CAPTCHAs, including Halligan (Teoh et al., 2025) and Oedipus (Deng et al., 2025). Although effective on specific CAPTCHA benchmarks, these systems frame CAPTCHA solving as a standalone problem, and their transferability to unseen CAPTCHA variants and broader GUI interaction remain limited.

To address this gap, we introduce ReCAP: a CAPTCHA-capable native GUI agent that can solve modern, interactive CAPTCHA challenges through its own GUI policy. Our approach is enabled by a dynamic, interactive CAPTCHA system spanning seven representative CAPTCHA types: Text, Compact Text, Icon Match, Icon Selection, Paged, Slider, and Image Grid. They are procedurally generated to capture diverse visual layouts and interaction patterns and to stress complementary CAPTCHA-solving capabilities. Building on this system, we develop a scalable data collection and curation pipeline that automatically generates large-scale successful interaction trajectories paired with chain-of-thought (CoT) reasoning traces. As CAPTCHA solving often requires multi-step interaction and recovery from intermediate mistakes, we further exploit failed trajectories to construct self-correction data, training agents to reflect on errors, adjust their reasoning, and revise actions toward the correct solution. To mitigate the imbalance caused by the large disparity in sequence lengths, we adopt a weighted training objective that balances reasoning and action tokens.

On held-out tests on our synthetic dynamic CAPTCHA benchmark, ReCAP-32B variants improve CAPTCHA solving success rates from roughly 30% to over 80% compared to the base agent. ReCAP also achieves substantial gain over its base agents on a real-world benchmark with 26 CAPTCHA variants, though transfer is mixed and frontier proprietary agents remain competitive on several categories. Finally, we present ablation studies and analyses that validate the contributions of CoT reasoning and self-correction data. By open-sourcing our dataset, model, and code, we aim to provide a foundation for future research on CAPTCHA-capable GUI agents.

## 2. Related Work

**GUI Agent** The development of Vision-Language Models (VLMs) has catalyzed a shift toward GUI agents capable of navigating graphical user interfaces. Early work in this domain primarily addressed grounding language instructions to static UI components using supervised learning on datasets of screen-action pairs. However, the field has rapidly transitioned toward end-to-end agents that operate on raw pixel observations to generate low-level control sequences (e.g., clicks, scrolls, and typing) (Nguyen et al., 2025; Zheng et al., 2024a). GUI agent benchmarks such as Mind2Web (Deng et al., 2023), ScreenSpot (Cheng et al., 2024), and Android Control (Li et al., 2024) have shifted the focus toward open-ended interaction across diverse operating systems and web environments.

Modern agents increasingly leverage high-resolution encoders and specialized architectures to handle fine-grained visual details. For instance, CogAgent (Hong et al., 2024) and Ferret-UI (You et al., 2024) introduce specialized vision towers to improve the perception of small UI elements. Recent work further improves GUI grounding through coordinate-free action heads, as in GUI-Actor (Wu et al., 2025a), and studies how visual attributes such as contrast, size, position, and card clarity influence web-agent decisions (Yu et al., 2026). Concurrently, the integration of structured reasoning has become a standard. Recent state-of-the-art models like UI-TARS (Qin et al., 2025) and OpenCUA (Wang et al., 2025) employ autoregressive formulations that interleave visual perception with chain-of-thought reasoning, while GUI-Libra (Yang et al., 2026) studies action-aware supervision and partially verifiable RL for reasoning-and-action training. This trend has been further reinforced by advances in multimodal foundation models, including large open-weight vision-language models that support direct action generation and tool use (Bai et al., 2025; Yang et al., 2025). Overall, the field is rapidly converging toward general-purpose GUI agents that can handle diverse interactive tasks using a unified perception-reasoning-action interface.

**Evolution of CAPTCHAs** CAPTCHAs were originally introduced as text-recognition challenges that exploited the gap between human perception and machine OCR capabilities (Ahn et al., 2003). More recently, reasoning CAPTCHAs mark a shift from simple pattern recognition toward tasks requiring logical reasoning, contextual understanding, and multi-step interactions (Gossweiler et al., 2009; Gao et al., 2021). Unlike traditional CAPTCHAs, their designs explicitly target cognitive abilities that remain difficult for automated solvers. As deep learning significantly improved OCR performance, text-based CAPTCHAs became largely ineffective, motivating a transition from "hard-to-read" to "hard-to-reason" challenges (Singh & Pal, 2014; Hitaj et al., 2020). Modern systems, such as Google's reCAPTCHA v2, employ rich visual-semantic tasks including image classification, object localization, and spatial reasoning (Searles et al., 2023). Beyond static puzzles, contemporary CAPTCHAs increasingly incorporate behavioral signals and multi-step interactive sequences to distinguish human-like interaction from automated scripts.

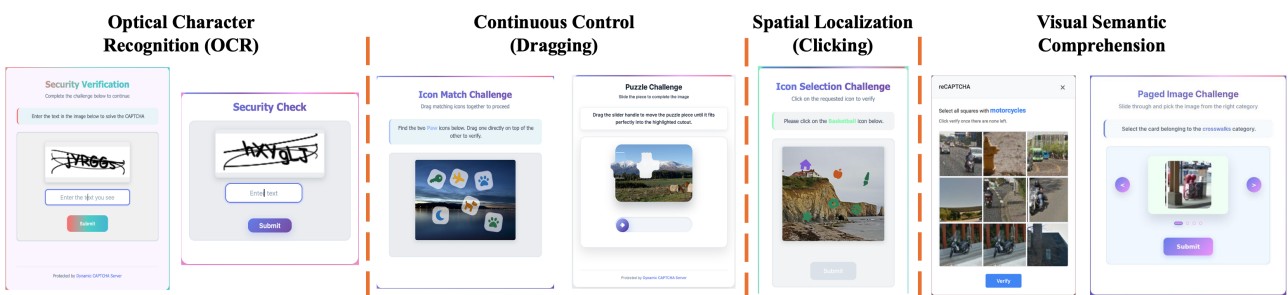

*Figure 2.* The suite of CAPTCHA challenges in our dynamic CAPTCHA system, designed to train fundamental CAPTCHA-solving primitives. The challenges are grouped into four core interaction primitives: Optical Character Recognition (OCR), Continuous Control (Dragging), Spatial Localization (Clicking), and Visual Semantic Comprehension.

**Automated CAPTCHA Solving** The evolution of CAPTCHA complexity has prompted a parallel advancement in automated solvers. Early approaches typically relied on task-specific pipelines. For instance, GeeSolver (Zhao et al., 2023) targets text CAPTCHAs with a specialized self-supervised learning pipeline, while Plesner et al. (2024) use YOLO-based segmentation and classification tailored to reCAPTCHA v2 image challenges. While effective, these methods lack the flexibility to adapt to new or hybrid CAPTCHA designs without significant retraining or redesign. More recent systems broaden the scope of automated CAPTCHA solving: PhishDecloaker (Teoh et al., 2024) uses a hybrid deep-vision interactive pipeline to solve challenges on CAPTCHA-cloaked phishing pages, while Halligan (Teoh et al., 2025) and Oedipus (Deng et al., 2025) explore VLM/LLM-based generalized or reasoning-oriented solvers. The Halligan framework formulates diverse visual CAPTCHA challenges as a unified search problem by mapping instructions to objectives and visual content to searchable representations. Oedipus targets reasoning CAPTCHAs by decomposing them into LLM-solvable substeps through a CAPTCHA-specific DSL. Complementarily, Reasoning under Vision (Song et al., 2025) studies visual-spatial cognition for CAPTCHA solving and shows that step-by-step reasoning can improve coordinate-grounded VLM performance.

Despite these gains, current generalized solvers often rely on repeated model calls, external heuristics, or explicit search procedures, which introduce high latency and fail to capture the temporal continuity of human interaction. There is a notable lack of research into whether GUI agents can natively internalize CAPTCHA-solving as a general capability.

## 3. Methodology

In this section, we outline the key components of our approach. We first introduce a dynamic CAPTCHA system that decomposes modern challenges into core interaction primitives. We then describe a data collection pipeline that produces solution and self-correction traces with explicit reasoning-action structure. Finally, we present a training paradigm that jointly models CoT reasoning and actions to integrate perception, reasoning, and interaction.

### 3.1. Dynamic CAPTCHA System

To train a successful CAPTCHA-solving agent, large-scale and diverse data are essential. Existing resources have substantially broadened CAPTCHA evaluation: Open Captcha-World (Luo et al., 2025) provides a web-based benchmark and platform with diverse dynamic CAPTCHA puzzles, while MCA-Bench (Wu et al., 2025b) offers a comprehensive and reproducible suite spanning heterogeneous CAPTCHA modalities. However, these resources are primarily designed for evaluation rather than closed-loop policy training, and therefore do not directly provide executable GUI feedback, full meta-supervision, or failure-recovery traces needed for scalable native-agent learning.

To bridge this gap, we conduct a comprehensive study based on prior literature Searles et al. (2023) and an extensive analysis of widely deployed CAPTCHAs including Google's reCAPTCHA, Geetest, Arkose Labs, hCAPTCHA, etc. From this study, we identify a set of reusable perception-action capabilities needed across different CAPTCHA types, which we refer to as *interaction primitives*, which are not consistently and properly emphasized in general GUI-agent datasets.

Motivated by these identified interaction primitives and their absence in existing datasets, we develop a dynamic CAPTCHA system spanning seven CAPTCHA challenges: Text, Compact Text, Icon Match, Icon Selection, Paged, Slider, and Image Grid. Each CAPTCHA challenge type is designed to model a set of core interaction primitives commonly observed in modern CAPTCHAs: optical character recognition (OCR), continuous control, precise spatial localization, and visual semantic comprehension. Collectively, these variants prepare the model with the core skills required to handle diverse CAPTCHA challenges. Figure 2 provides an overview of the challenge variants in our dynamic CAPTCHA system. Each challenge is randomly generated and accompanied by a meta API that provides access to complete groundtruth information, including answer label, target coordinates, user interactions, etc. The environment supports interactive actions with real-time feedback.

Upon submission, the server returns structured JSON feedback (e.g., *solved: true/false*). More detailed description of the CAPTCHA challenges and their design is provided in Appendix Section C.

**Stochastic Rendering**  A key limitation of specialized datasets is their tendency to induce overfitting to fixed layout artifacts. To address this, our Flask-based rendering engine injects stochasticity into many layers of the DOM at request time. The server dynamically samples CSS variables controlling spatial layout (e.g., padding, margins, container widths, etc.) and styling (e.g., color palettes, font sizes, weights, etc.), in addition to alternating submission methods. This deliberate visual and structural variance encourages the agent to rely on semantic understanding rather than fixed color or layout heuristics.

**Unbounded Generation via Visual Diversity**  In addition to stochastic layout and styling, our CAPTCHA system includes a broad bank of visual assets, including icons from Font Awesome, categorized images (Ma, 2024), background images (Dung, 2022), and distorted text images (hammer888, 2024). These assets are randomly composed with varying colors, scales, and placements, yielding a combinational space that allows the system to generate effectively unbounded unique CAPTCHA instances. This diversity mitigates overfitting to specific visual patterns and promotes robust generalization across appearance variations.

Overall, the proposed system combines task-specific challenge designs with stochastic rendering and visual diversity to generate an effectively unbounded CAPTCHA space, encouraging semantic reasoning over layout memorization and improving robustness to real-world variations.

### 3.2. Scalable Data Collection and Curation

With the help of our dynamic CAPTCHA system, generating large-scale CAPTCHA trajectory data becomes possible. The ground-truth annotations (e.g., target coordinates and environment metadata) provided by our system enable scalable generation of expert solutions and self-correction traces, as we will discuss in this section.

**Solution Trace Generation with Reasoning Data**  One key challenge in improving CAPTCHA-solving capability is collecting large-scale training data with minimal human supervision. To address this challenge, we utilize the *groundtruth* provided by our dynamic CAPTCHA system. This supervision guides an expert reasoning model (e.g., a SOTA strong model) toward producing both correct final answers and aligned intermediate reasoning steps. Specifically, we provide the expert with a screenshot of the CAPTCHA and a carefully curated prompt that prompts the expert to reason as if the ground truth were unknown. However, we also explicitly supply the ground truth solution and annotation, which constrains the expert's final action sequence to

match the ground truth. The expert generates structured reasoning, and we perform automatic checks to ensure that the reasoning trace does not cheat and directly reveal the answer. We append the ground truth actions and obtain reasoning-action traces. This answer-conditioned setup is similar in spirit to rationale bootstrapping in STaR (Zelikman et al., 2022). Manual inspection of sampled traces found them generally coherent and task-aligned. Figure 3(a) illustrates the workflow of solution trace generation.

The reasoning-action trace offers several advantages over action-only supervision. Explicit reasoning exposes intermediate semantic decisions such as object identification, target verification, and termination conditions, allowing the model to learn not only what action to take but why it is taken. By abstracting away from pixel-level appearance, reasoning provides a consistent decision structure across visually diverse CAPTCHA instances, which in turn improves generalization and enables reliable transfer of learned primitives to unseen variants and out-of-distribution layouts. In Section 4.5, we present an ablation study to validate the importance of reasoning data.

**Self-correction Trace Generation**  In addition to expert solution traces, we generate self-correction traces through a rejection-sampling-based pipeline that targets realistic model failures. Self-correction traces expose the model to realistic failure cases that are not captured by expert solution traces alone. By retrospectively analyzing incorrect attempts and demonstrating how erroneous reasoning or actions should be revised, these traces provide explicit supervision for error recovery and help the model avoid repeating common mistakes on challenging variants. We validate their importance through an ablation study in Section 4.6.

Figure 3(b) illustrates the workflow of the self-correction trace generation pipeline. For each CAPTCHA instance, we first use a student model to attempt solving the challenge and record its full interaction trace, including intermediate reasoning and actions. The system then automatically verifies the results against ground truth. If the solution is incorrect, we treat the attempt as a valuable learning signal and trigger the generation of the self-correction trace. We provide the expert model with the complete context of the unsuccessful attempt. This includes the original and intermediate screenshots and the student model's reasoning-action trace. Using a specially curated prompt, the expert model is instructed to retrospectively analyze the failure. The expert model is asked to identify the source of the error and generate a corrective reasoning trace as if it had discovered the mistake autonomously, while implicitly guided by the recovery ground truth solution and annotations.

**Multi-Action Outputs**  Unlike many GUI tasks, one unique challenge in CAPTCHA systems is their imposed time limitation. Most GUI agents that emit a single action

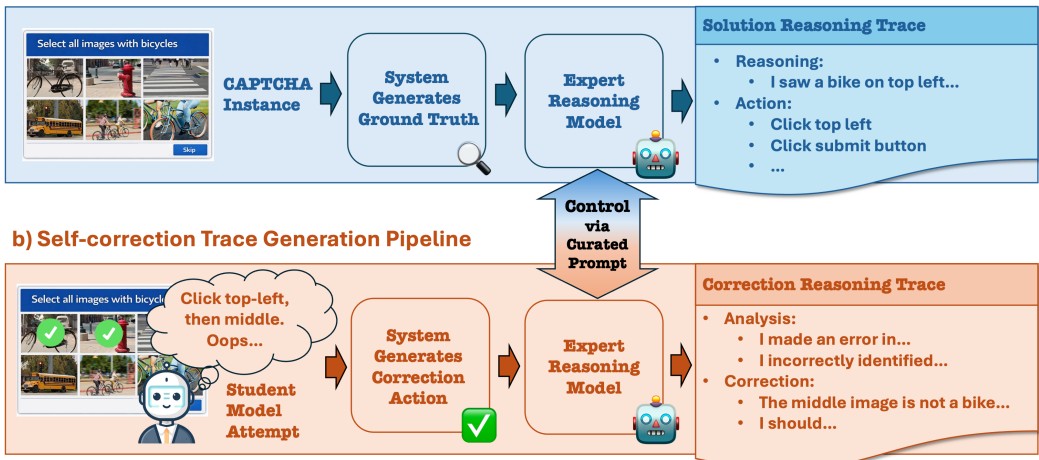

**a) Reasoning Solution Trace Generation Pipeline**

**b) Self-correction Trace Generation Pipeline**

*Figure 3.* Data collection and curation pipeline for our CAPTCHA training dataset. a) shows the reasoning solution trace generation pipeline; b) shows the self-correction trace generation pipeline.

per step are unsuitable for solving complex CAPTCHA. Our data contain structured output sequences with multiple GUI actions within a single response when necessary (e.g., sequential clicks required to solve a CAPTCHA). This design reduces unnecessary interaction rounds, important for solving real-world CAPTCHA systems.

### 3.3. Training Paradigm

Our scalable reasoning and self-correction data curation enables us to finetune state-of-the-art (SOTA) VLMs for CAPTCHA capability. Each training sample consists of an interleaved sequence of reasoning text and structured action tokens, where a single response may contain multiple GUI actions. As reasoning traces are typically much longer than action sequences, naive token-level training can overemphasize linguistic generation at the expense of precise interaction. To address this imbalance, we reweigh reasoning and action tokens within a unified loss function.

**Unified Loss Function** Given a training trajectory represented as a token sequence $\mathbf{y}$, we define two disjoint index sets: $\mathcal{T}$ for reasoning tokens and $\mathcal{A}$ for action tokens. The model is trained autoregressively to predict each token conditioned on the visual observation $\mathbf{x}$ and all preceding tokens. We then use a unified negative log-likelihood objective with group-level weights assigned to reasoning and action tokens:

$$\mathcal{L}_{\text{total}} = -\Big( \frac{\lambda_{\text{think}}}{|\mathcal{T}|} \sum_{t \in \mathcal{T}} \log p_\theta(y_t \mid \mathbf{x}, y_{<t}) + \frac{\lambda_{\text{act}}}{|\mathcal{A}|} \sum_{t \in \mathcal{A}} \log p_\theta(y_t \mid \mathbf{x}, y_{<t}) \Big).$$

where $\lambda_{\text{think}}$ and $\lambda_{\text{act}}$ are tunable hyperparameters. This unified loss preserves the simplicity of standard model training while preventing long reasoning traces from dominating

short action sequences.

## 4. Experiments

In this section, we report the model training setup and statistics we obtain from the experiments and ablation studies.

### 4.1. Model Training

We follow the training paradigm described in Section 3.3 to finetune Qwen3-VL-8B-Thinking and Qwen3-VL-32B-Thinking (Yang et al., 2025). We additionally finetune Qwen3-VL-8B-Instruct and Qwen3-VL-32B-Instruct with the same data and objective. For the Thinking models, we retain the native Qwen3-VL reasoning format and supervise CoT traces directly before the action tokens. For the Instruct models, we wrap CoT traces with `<thinking>` tags while keeping all other training settings unchanged. Both the reasoning and action losses are weighted equally, with $\lambda_{\text{think}} = \lambda_{\text{act}} = 0.5$, and models are trained for a single epoch over approximately 230,000 samples. Additional training configuration is provided in Appendix Section E.3. We refer to the resulting models as ReCAP-8B (Thinking), ReCAP-32B (Thinking), ReCAP-8B (Instruct), and ReCAP-32B (Instruct).

**Training Data** Our training corpus consists of approximately 150,000 solution trajectories and 10,000 self-correction trajectories generated by our scalable data collection and curation pipeline (Section 3.2) with Qwen2.5-VL-72B-Instruct as the expert reasoning model, covering all seven CAPTCHA challenge types evenly. To preserve general GUI interaction capabilities, we additionally mix in approximately 50,000 general GUI grounding and interaction trajectories from Aguvis (Xu et al., 2024) and 23,000 interaction trajectories from AgentNet (Wang et al., 2025). This combined training set enables the model to

*Table 1.* **CAPTCHA solving performance on our dynamic CAPTCHA system** across different models and frameworks. All models are evaluated on a fixed set of 1,000 CAPTCHA instances generated by the same dynamic rendering engine, using a disjoint held-out testing set. SR (%) denotes the solve rate; Avg. Steps is calculated based on the average number of model calls in successful test cases. Higher SR (↑) and lower steps (↓) are better. Best values are **highlighted** and second-best values are underlined per column. Step counts are omitted where unavailable.

| Model | Metric | Text | Compact Text | Icon Match | Icon Selection | Paged | Slider | Image Grid | Overall |
|---|---|---|---|---|---|---|---|---|---|
| *Open-source GUI Agents* | | | | | | | | | |
| UI-TARS-1.5-7B | SR (%) ↑ | 28.57 | 17.36 | 51.30 | 66.01 | 44.68 | 18.88 | 0.75 | 33.60 |
| | Avg. Steps ↓ | 2.31 | 2.39 | 2.24 | 1.78 | 2.98 | 2.56 | 3.44 | 2.47 |
| Qwen3-VL-8B-Thinking | SR (%) ↑ | 33.54 | 25.83 | 27.49 | 41.10 | 59.85 | 5.47 | 4.38 | 28.70 |
| | Avg. Steps ↓ | 2.44 | 2.55 | 2.66 | 1.83 | 2.84 | 2.71 | 4.00 | 2.52 |
| Qwen3-VL-32B-Thinking | SR (%) ↑ | 37.89 | 29.17 | 25.15 | 38.36 | 63.50 | 8.59 | 2.92 | 29.70 |
| | Avg. Steps ↓ | 2.51 | 2.57 | 2.74 | 1.58 | 2.99 | 3.00 | 3.25 | 2.55 |
| Qwen3-VL-8B-Instruct | SR (%) ↑ | 48.34 | 45.77 | 0.00 | 65.44 | 13.70 | 0.00 | 0.00 | 24.70 |
| | Avg. Steps ↓ | 3.00 | 3.00 | – | 1.39 | 2.95 | – | – | 2.51 |
| Qwen3-VL-32B-Instruct | SR (%) ↑ | 35.76 | 32.39 | 64.08 | 71.32 | 50.68 | 13.99 | 7.86 | 39.30 |
| | Avg. Steps ↓ | 2.99 | 3.00 | 1.75 | 1.60 | 2.55 | 3.20 | 3.91 | 2.31 |
| *Closed-source GUI Agents* | | | | | | | | | |
| OpenAI CUA | SR (%) ↑ | 36.02 | 33.33 | 0.00 | 81.50 | 65.69 | 0.00 | 8.03 | 31.80 |
| | Avg. Steps ↓ | 3.00 | 2.95 | – | 1.57 | 2.61 | – | 3.82 | 2.33 |
| GPT-5.4 | SR (%) ↑ | 58.94 | 55.63 | 92.25 | 94.85 | 30.82 | 37.76 | 29.29 | 56.80 |
| | Avg. Steps ↓ | 2.08 | 2.08 | 2.02 | 2.21 | 2.56 | 2.39 | 3.00 | 2.23 |
| Gemini-3-Flash-Preview | SR (%) ↑ | 62.91 | **71.83** | 89.44 | 81.62 | 31.51 | 79.72 | 10.71 | 59.90 |
| | Avg. Steps ↓ | 1.67 | 1.72 | 1.74 | 2.42 | 2.90 | 1.79 | 3.00 | 1.98 |
| *Specialized Pipelines* | | | | | | | | | |
| Halligan Framework | SR (%) ↑ | 42.00 | 44.00 | 2.00 | 14.00 | 8.00 | 2.00 | 64.00 | 25.14 |
| | Avg. Steps ↓ | ≥ 3 | ≥ 3 | ≥ 5 | ≥ 3 | ≥ 5 | ≥ 3 | ≥ 3 | – |
| *Ours* | | | | | | | | | |
| ReCAP-8B (Thinking) | SR (%) ↑ | 60.25 | 47.50 | 95.32 | 80.14 | 72.99 | 88.28 | 52.55 | 71.90 |
| | Avg. Steps ↓ | 1.24 | 1.25 | 1.14 | 1.38 | 2.59 | 1.42 | 2.11 | 1.54 |
| ReCAP-32B (Thinking) | SR (%) ↑ | 62.11 | 55.83 | **99.42** | 82.88 | **85.40** | **100.00** | **78.10** | **81.00** |
| | Avg. Steps ↓ | 1.31 | 1.34 | 1.03 | 1.12 | 2.87 | **1.00** | 2.21 | 1.54 |
| ReCAP-8B (Instruct) | SR (%) ↑ | 65.56 | 61.27 | 97.89 | 92.65 | 75.34 | 88.11 | 70.71 | 78.60 |
| | Avg. Steps ↓ | 1.03 | **1.02** | 1.03 | 1.02 | **2.26** | **1.00** | **2.10** | **1.33** |
| ReCAP-32B (Instruct) | SR (%) ↑ | **80.13** | 66.20 | 97.89 | **95.59** | 73.29 | **100.00** | 77.14 | **84.20** |
| | Avg. Steps ↓ | **1.00** | **1.02** | **1.01** | **1.01** | 2.46 | **1.00** | 2.44 | 1.38 |

acquire CAPTCHA-specific interaction skills while maintaining performance on broader tasks.

## 4.2. Evaluation on Dynamic CAPTCHA System

### 4.2.1. EXPERIMENT SETUP

We evaluate a range of open-source GUI agent models, including UI-TARS-1.5-7B (Qin et al., 2025), Qwen3-VL-8B-Thinking, Qwen3-VL-32B-Thinking, Qwen3-VL-8B-Instruct and Qwen3-VL-32B-Instruct, as well as closed-source GUI agents including OpenAI's Computer-Use Agent (OpenAI CUA), GPT-5.4, and Gemini-3-Flash-Preview. In addition, we benchmark Halligan (Teoh et al., 2025) as a state-of-the-art specialized CAPTCHA-solving framework. Finally, we benchmark four ReCAP variants: ReCAP-8B (Thinking), ReCAP-32B (Thinking), ReCAP-8B (Instruct), and ReCAP-32B (Instruct).

All models are evaluated on a fixed set of 1,000 CAPTCHA

instances generated by the same dynamic rendering engine, using a **disjoint held-out testing set**. The experiment covers all seven challenges evenly with substantial variation in layout, styling, and appearance. For each CAPTCHA challenge, models are given a budget of 5 steps to solve the challenge, while Image Grid and Paged challenges are given 8 steps. For the Halligan framework, we evaluate each CAPTCHA type on 100 samples due to budget constraints, as evaluation requires a large number of OpenAI API calls and does not impose an interaction-step limit.

Our primary evaluation metric is the CAPTCHA solving success rate, defined as the fraction of challenges successfully solved within the interaction budget. We additionally report the average number of model calls in successful test cases. Together, these metrics capture both end-to-end task success and the efficiency of GUI interactions.

### 4.2.2. RESULTS & DISCUSSION

Table 1 summarizes the performance of all evaluated models and frameworks. **ReCAP model variants significantly outperform all baselines in overall CAPTCHA-solving success rates.** Most notably, ReCAP-32B (Instruct) achieves the highest overall success rate, improving over Qwen3-VL-32B-Instruct from 39.30% to 84.20%. All ReCAP variants also outperform frontier GUI-agent baselines GPT-5.4 (56.80%) and Gemini-3-Flash-Preview (59.90%) overall, although these models remain competitive on individual categories such as Icon Match, Icon Selection, Compact Text, and Slider. These results show that reasoning-action training can turn Qwen3-VL models into strong native CAPTCHA solvers. In contrast, general-purpose GUI agents remain much weaker on CAPTCHA-specific interaction patterns, especially tasks that require robust OCR under distortion, precise spatial reasoning, and continuous control. The Instruct variants further improve over the corresponding Thinking variants, while scaling from 8B to 32B generally improves robustness for the Thinking variants and on several visually complex challenge types.

**ReCAP models are significantly more efficient.** The ReCAP variants achieve substantially higher solve rates than general GUI agents and specialized pipelines while using fewer model-call steps on successful cases. For example, ReCAP-8B (Instruct) requires only an average of 1.33 model-call steps, lower than Gemini-3-Flash-Preview (1.98), GPT-5.4 (2.23), OpenAI CUA (2.33), the open-source GUI-agent baselines (2.47–2.55), and the Halligan framework (3-5). It is important to note that Halligan step numbers represent the minimum interaction steps and are often higher in practice due to multiple sequential API calls. Since each step involves model inference, GUI state recording, and action execution, step count is a useful proxy for interaction overhead and timeout risk, although we do not provide a full latency decomposition. In practice, pipelines with numerous interaction rounds may be more likely to exceed CAPTCHA time limits or fail due to refresh or expiration. ReCAP models reduce the number of required interaction rounds, which is an important source of practical efficiency.

### 4.3. Evaluation on Real-World CAPTCHAs

#### 4.3.1. EXPERIMENT SETUP

To evaluate whether the skills learned from our dynamic CAPTCHA system can be transferred to CAPTCHAs in the wild, we further benchmark our models on real-world CAPTCHA challenges following the evaluation protocol and dataset introduced by Halligan (Teoh et al., 2025). This benchmark includes interactive visual CAPTCHAs collected from major CAPTCHA providers, covering a wide range of challenge designs and interaction patterns encountered in the real world. Specifically, the benchmark includes CAPTCHAs from Google's reCAPTCHA, hCAPTCHA,

*Table 2.* **CAPTCHA solving performance on real-world CAPTCHAs** across different models and frameworks. Our models are evaluated in a *zero-shot* setting without training on this real-world CAPTCHA dataset, whereas Halligan (H) is *explicitly designed* for these CAPTCHA types. **ReCAP-8B (I)** and **ReCAP-32B (I)** denote ReCAP-8B (Instruct) and ReCAP-32B (Instruct), respectively. **H** denotes the Halligan framework. **Qwen-8B (I)** and **Qwen-32B (I)** denote Qwen3-VL-8B-Instruct and Qwen3-VL-32B-Instruct, respectively. All numbers are solve rates (%). Best values are **highlighted** and second-best values are underlined per row.

| CAPTCHA Type | ReCAP -8B (I) | ReCAP -32B (I) | H | Qwen -8B (I) | Qwen -32B (I) |
|---|---|---|---|---|---|
| tencent/vtt | 24 | 39 | 8 | **42** | 37 |
| mtcaptcha | 65 | **73** | 66 | 63 | 56 |
| yandex/text | 74 | **85** | 82 | 81 | 80 |
| botdetect | **83** | 72 | 80 | 74 | 78 |
| recaptchav2 | 68 | **74** | 61 | 12 | 36 |
| baidu/rotate | 21 | 22 | **79** | 0 | 0 |
| hcaptcha | 47 | **63** | 2 | 37 | 62 |
| geetest/slide | 68 | **80** | 16 | 0 | 16 |
| lemin | **18** | 12 | 0 | 0 | 0 |
| amazon/waf | 20 | **27** | 6 | 15 | 23 |
| funcaptcha/counting | 29 | 36 | **54** | 32 | 33 |
| funcaptcha/hand_number | 34 | 26 | **40** | 32 | 34 |
| funcaptcha/galaxies | 99 | **100** | **100** | **100** | 99 |
| funcaptcha/dice_pair | 30 | **35** | 27 | 33 | 20 |
| funcaptcha/card | 33 | 24 | 25 | **38** | 31 |
| funcaptcha/square_icon | 46 | 54 | **82** | 63 | 73 |
| funcaptcha/rotated | 82 | **85** | **85** | 79 | 74 |
| arkose/3d_rollball | **50** | 38 | 1 | 0 | 3 |
| arkose/dice_match | **51** | 37 | 8 | 0 | 0 |
| arkose/orbit_match | **38** | 30 | 8 | 0 | 7 |
| arkose/rockstack | **57** | 33 | 13 | 4 | 14 |
| arkose/numbermatch | **43** | 39 | 8 | 3 | 9 |
| geetest/icon | 15 | 15 | 2 | 6 | **19** |
| geetest/iconcrush | 6 | 8 | **70** | 14 | 5 |
| geetest/gobang | 0 | 0 | **36** | 4 | 1 |
| yandex/kaleidoscope | 8 | 3 | **47** | 0 | 3 |

GeeTest, Arkose Labs, and Amazon, spanning diverse task formats including image selection, icon matching, rotation alignment, object counting, and slider-based puzzles.

In the main comparison, we evaluate Qwen3-VL-8B-Instruct, Qwen3-VL-32B-Instruct, the corresponding ReCAP Instruct variants, and the Halligan framework. Following Teoh et al. (2025), all challenges are evaluated in an interactive setting where the model observes rendered CAPTCHA frames, issues GUI actions, and receives binary success feedback upon submission. A challenge is considered successfully solved if the model completes the task and obtains a valid pass signal within the allowed interaction budget.

Importantly, our models are *not trained* on this real-world CAPTCHA dataset and are evaluated in a *zero-shot transfer* setting, whereas Halligan is *explicitly designed* to handle these specific CAPTCHA types. For a fair comparison,

*Table 3.* **Model performance on general GUI agent benchmarks**. Columns are grouped by Thinking and Instruct variants. Android Control, ScreenSpot-V2, Multimodal-Mind2Web, and UI-Vision Element report success rates or accuracies (%). UI-Vision Element uses the overall element-grounding accuracy from UI-Vision; UI-Vision Layout reports mean IoU for layout grounding. Best values are **highlighted** and second-best values are underlined separately within the Thinking and Instruct groups for each row.

| Benchmark | Thinking variants | | | | Instruct variants | | | |
|---|---|---|---|---|---|---|---|---|
| | ReCAP -8B | ReCAP -32B | Qwen -8B | Qwen -32B | ReCAP -8B | ReCAP -32B | Qwen -8B | Qwen -32B |
| Android Control (low) | 58.60 | **67.40** | 65.80 | 67.20 | 62.60 | **67.40** | 66.60 | 67.00 |
| Android Control (high) | 43.80 | 49.80 | 49.00 | **50.80** | 48.00 | **50.20** | 50.00 | 49.80 |
| ScreenSpot-V2 | 80.03 | 93.24 | 92.06 | **94.50** | 89.54 | **91.90** | 91.12 | 92.80 |
| Multimodal-Mind2Web | 38.37 | **44.73** | 42.58 | 42.91 | 48.18 | **49.57** | 46.52 | 47.10 |
| UI-Vision (Element) | 7.78 | 12.81 | 7.48 | **13.20** | 22.05 | 36.56 | 29.72 | **37.76** |
| UI-Vision (Layout) | 30.50 | **49.80** | 33.10 | 48.80 | 45.10 | **55.66** | 42.98 | 48.43 |

all Halligan results are obtained by running the released framework under the same evaluation protocol.

### 4.3.2. RESULTS & DISCUSSION

Table 2 presents the results. **ReCAP models consistently outperform the corresponding baseline on most real-world CAPTCHA types, indicating effective skill transfer to CAPTCHAs in the wild.** Averaged across all 26 CAPTCHA types, ReCAP-8B (Instruct) and ReCAP-32B (Instruct) improve over their corresponding Qwen3-VL baselines by 14.50 pp and 11.42 pp, respectively, and both outperform Halligan by a small margin on average. The ReCAP Instruct variants are especially strong on interaction-heavy challenges such as *recaptchav2*, *geetest/slide*, and multiple *Arkose Lab* variants, despite not being trained on the benchmark. A similar trend is observed for the Thinking variants; comparisons with GPT-5.4 and Gemini-3-Flash-Preview are mixed across CAPTCHA types. We provide the full per-type breakdown and discussion in Appendix Section E.2. Scaling from 8B to 32B produces mixed gains. While ReCAP-32B (Instruct) provides clear benefits on several visually complex challenges, ReCAP-8B (Instruct) remains competitive and occasionally performs better on structured challenges such as *botdetect* and several *Arkose Lab* variants.

The Halligan framework remains noticeably stronger on some rotation and pattern-recognition CAPTCHAs (e.g., *baidu/rotate*, *geetest/iconcrush*, and *yandex/kaleidoscope*), highlighting the complementary strengths of task-specific frameworks. However, Halligan typically requires substantially longer interaction sequences and higher latency, whereas our models can solve many challenges within a small number of interaction steps.

### 4.4. Evaluation on General GUI Agent Benchmarks

**Experiment Setup** In addition to CAPTCHA-specific evaluations, we assess whether finetuning on our dynamic CAPTCHA system preserves general GUI interaction capabilities. To this end, we evaluate our models on established general-purpose GUI agent benchmarks, including Android Control (Li et al., 2024), ScreenSpot-V2 (Cheng

et al., 2024), Multimodal-Mind2Web (Zheng et al., 2024b; Deng et al., 2023), and UI-Vision (Nayak et al., 2025). The benchmark introduction and evaluation settings are provided in Section D.3.

We evaluate the Qwen3-VL baselines and the corresponding ReCAP variants described in Section 4.2.1. All evaluations follow the official benchmark protocols, metrics, and task splits. Performance is reported using each benchmark's standard metrics. These experiments are intended to examine how CAPTCHA-oriented finetuning affects general GUI grounding and interaction ability.

**Results & Discussion** Table 3 groups the results by Thinking and Instruct variants. For the Thinking variants, ReCAP-32B largely preserves general GUI performance: it slightly improves over Qwen3-VL-32B-Thinking on Android Control (low) and Multimodal-Mind2Web, remains close on Android Control (high), ScreenSpot-V2, and UI-Vision Element, and improves UI-Vision Layout IoU. This suggests that the 32B Thinking model can absorb CAPTCHA-specific reasoning-action supervision while retaining broad GUI grounding ability.

The Instruct variants show a similar but more task-dependent pattern. ReCAP-32B (Instruct) remains close to Qwen3-VL-32B-Instruct on ScreenSpot-V2 and UI-Vision Element, while substantially improving UI-Vision Layout IoU from 0.4843 to 0.5566. ReCAP-8B (Instruct), however, drops on ScreenSpot-V2 and UI-Vision Element, although it improves UI-Vision Layout IoU over its baseline. Overall, the 8B variants are more vulnerable to degradation after CAPTCHA finetuning, whereas the 32B variants better preserve general GUI capabilities and can even improve layout grounding. We attribute this trend to model capacity: jointly learning CAPTCHA-specific interaction primitives and general GUI behaviors introduces greater representational interference for smaller models.

### 4.5. Ablation Study on CoT Reasoning

We evaluate the impact of CoT reasoning traces by comparing the Qwen3-VL-8B-Thinking model finetuned with

and without explicit CoT reasoning supervision. Both models are trained on the same data, and the only difference is whether reasoning traces are included during training.

### 4.5.1. EXPERIMENT SETUP

The experiment setup follows Section 4.2.1. We benchmark both models on 1,000 CAPTCHA challenges from our dynamic CAPTCHA system. Each model is allocated a maximum of 5 interaction steps per challenge, while Image Grid and Paged challenges are allocated up to 8 steps due to their higher interaction complexity.

### 4.5.2. RESULTS AND DISCUSSION

*Table 4.* **Ablation study evaluating the impact of CoT reasoning on CAPTCHA solving performance.** The table compares Qwen3-VL-Thinking-8B model finetuned without CoT reasoning data against the same base model finetuned with CoT reasoning data across seven CAPTCHA challenges. SR (%) denotes the success rate; Avg. Steps is computed over successful test cases.

| CAPTCHA Type | w/o Reasoning Data | | w/ Reasoning Data | |
|---|---|---|---|---|
| | SR (%) | Avg. Steps | SR (%) | Avg. Steps |
| Text | 55.90 | 1.33 | 60.25 | 1.24 |
| Compact Text | 45.00 | 1.36 | 47.50 | 1.25 |
| Icon Match | 88.89 | 1.16 | 95.32 | 1.14 |
| Icon Selection | 74.66 | 1.49 | 80.14 | 1.38 |
| Paged | 69.34 | 2.69 | 72.99 | 2.59 |
| Slider | 84.38 | 1.44 | 88.28 | 1.42 |
| Image Grid | 40.88 | 2.20 | 52.55 | 2.11 |
| **Overall** | **66.40** | **1.61** | **71.90** | **1.54** |

Table 4 shows that adding reasoning traces consistently improves performance across all CAPTCHA types, increasing the overall success rate from 66.40% to 71.90%. The largest gains occur on complex tasks such as Image Grid and Icon Match. Average steps remain comparable (1.61 → 1.54), indicating that reasoning traces improve decision quality rather than interaction length. **The findings suggest that explicit CoT reasoning supervision enhances CAPTCHA-solving accuracy without increasing the number of interaction steps.**

### 4.6. Ablation Study on Self-Correction Trace

We also evaluate the impact of self-correction traces by comparing Qwen3-VL-8B-Thinking model finetuned with and without self-correction traces. Both models are trained with full reasoning-action supervision on expert solution traces. The only difference is the inclusion of self-correction traces derived from failed attempts.

### 4.6.1. EXPERIMENT SETUP

This ablation study follows the same setup described in Section 4.5.1.

*Table 5.* **Ablation study evaluating the impact of self-correction traces on CAPTCHA solving performance.** The table compares Qwen3-VL-Thinking-8B model finetuned without self-correction traces against the same base model finetuned with self-correction traces across seven CAPTCHA challenges. SR (%) denotes the success rate; Avg. Steps is computed over successful test cases.

| CAPTCHA Type | w/o Correction Traces | | w/ Correction Traces | |
|---|---|---|---|---|
| | SR (%) | Avg. Steps | SR (%) | Avg. Steps |
| Text | 43.48 | 1.04 | 60.25 | 1.24 |
| Compact Text | 40.00 | 1.17 | 47.50 | 1.25 |
| Icon Match | 86.55 | 1.26 | 95.32 | 1.14 |
| Icon Selection | 78.08 | 1.39 | 80.14 | 1.38 |
| Paged | 69.34 | 2.63 | 72.99 | 2.59 |
| Slider | 66.41 | 1.55 | 88.28 | 1.42 |
| Image Grid | 50.36 | 2.17 | 52.55 | 2.11 |
| **Overall** | **62.90** | **1.60** | **71.90** | **1.54** |

### 4.6.2. RESULTS AND DISCUSSION

Table 5 presents the results. Self-correction traces yield consistent performance gains across all seven CAPTCHA types. Improvements are especially pronounced on text-heavy challenges such as Text (+16.77pp) and Compact Text (+7.50pp), where recognition errors are common. Interaction-intensive tasks that require precise control and multi-step coordination, including Slider (+21.9pp), Icon Match (+8.77pp), and Paged (+3.65pp), also benefit noticeably. Even on Image Grid, where the task is already challenging, we observe a modest improvement (+2.19pp). The average number of steps remains comparable (overall 1.60 → 1.54), indicating that the model becomes more successful without requiring substantially longer interaction sequences.

Overall, the success rate improves from 62.90% to 71.90%. **These findings validate self-correction traces as a key component of the data curation pipeline and substantially improve robustness across all CAPTCHA types.**

## 5. Conclusion

We present ReCAP, a CAPTCHA-capable native GUI agent that solves modern, interactive CAPTCHA challenges. By introducing a dynamic CAPTCHA system and a scalable reasoning-action data generation pipeline with retrospective self-correction traces, ReCAP learns the core interaction primitives underlying contemporary CAPTCHAs. Experiments show that models trained under this paradigm achieve strong performance on held-out dynamic CAPTCHAs, useful zero-shot transfer to real-world CAPTCHA benchmarks, and fewer interaction steps, while largely preserving general GUI agent performance. These results position CAPTCHA solving not as a standalone task, but as a transferable skill set that advances the capabilities of native GUI agents.

## Acknowledgment

The authors acknowledge the support of the AI Safety Fund administered by the Meridian Institute and the Frontier Model Forum (FMF). This work was also in part supported by NSF grants 2229876.

## Impact Statement

While our results demonstrate that native GUI agents can solve a broad class of modern CAPTCHAs, the goal of this work is not to enable real-world exploitation, but to probe the limits of vision-language model reasoning under adversarial, interactive conditions. By systematically decomposing CAPTCHA challenges into a set of learnable interaction primitives and analyzing agentic self-correction behaviors, we provide a structured framework for stress-testing existing human-verification mechanisms. These insights are intended to boost the development of more robust verification systems, motivating a shift away from static visual puzzles toward behavior-centric protocols that are more resilient to emerging GUI agents.

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

## A. Limitations of this Study

Like many similar studies, we rely on synthetic data to train our model. While this allows us to generate a large number of examples, there remains a gap between these simulations and the messy, unpredictable real world. Our strongest empirical claim is therefore specific to the held-out dynamic CAPTCHA benchmark, while real-world transfer remains mixed.

Second, we focus on the logical and visual-interaction aspects of solving CAPTCHA puzzles, not on mimicking human behavior. We do not simulate factors like mouse speed, hesitation, cursor acceleration, or other behavioral biometrics, which some security systems use to detect bots even if the puzzle is solved correctly.

Finally, our reasoning traces are answer-conditioned and should be interpreted as scalable training supervision rather than verified faithful explanations. We also leave several evaluation extensions to future work, including an Oedipus comparison, full latency decomposition, and a systematic sweep over alternative reasoning/action loss weights.

## B. Self-correction Capability Case Study

This section presents qualitative case studies that illustrate the model's self-correction capability under realistic CAPTCHA-solving scenarios. We focus on two representative challenge types: an interaction-intensive image-grid challenge and a classic text-like challenge. In all cases, we include reasoning traces and highlight the intermediate decision-making process that leads to successful correction. For clarity, we omit the low-level GUI action sequences in this section, as they follow directly from the reasoning.

### B.1. Image Grid Challenge

**User**

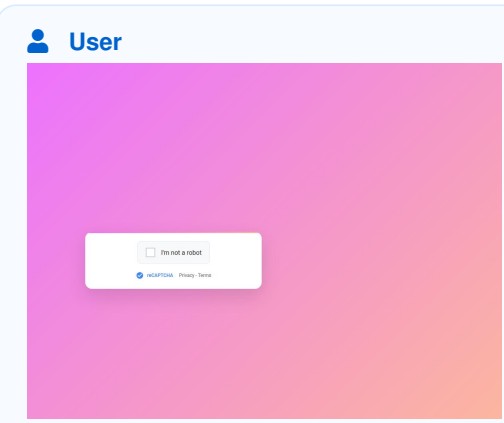

Solve the CAPTCHA as per the given instructions. You may need to interact with various elements such as checkboxes, image grids, sliders, or text inputs to complete the CAPTCHA challenge.

**Assistant**

I've encountered a reCAPTCHA checkbox verification. My procedure will be to click the box, await the loading of the image matrix, and finally, select the tiles as prompted.

**👤 User**

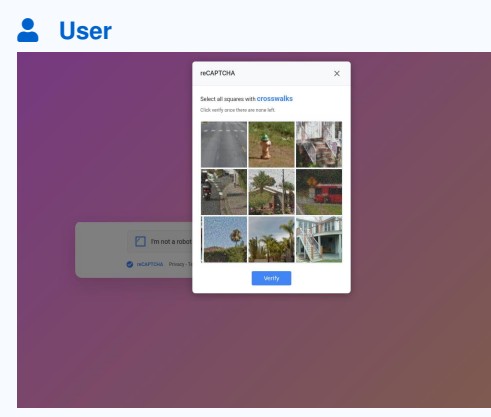

Continue solving the CAPTCHA. Observe the current state and take the next actions.

**🤖 Assistant**

I observe an image grid with nine image tiles, none of which depict crosswalks. The images show various outdoor scenes including roads, a fire hydrant, a bicycle, and buildings. I infer that the task requires selecting tiles with crosswalks, but since none are present, no tiles need to be selected. I plan to open the image grid, confirm the absence of crosswalks in all tiles, and then press the 'Verify' button to complete the challenge as instructed.

**👤 User**

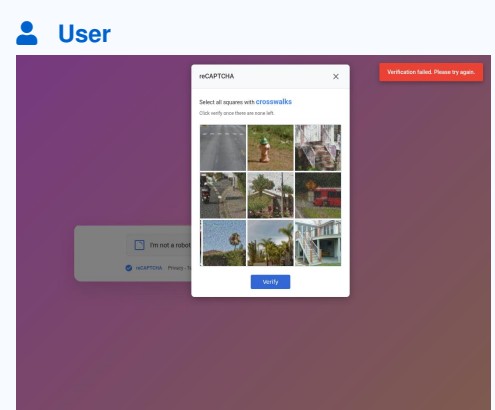

Continue solving the CAPTCHA. Observe the current state and take the next actions.

**🤖 Assistant**

I observe a reCAPTCHA grid with nine image tiles, each depicting various outdoor scenes including roads, a fire hydrant, a bicycle, and buildings. The instruction asks to select all squares with crosswalks. Upon careful examination, tile 0 does indeed show a crosswalk, while the others do not. The correct approach is to carefully identify and select only the tile with crosswalks. I plan to click on tile 0, as it matches the criteria, and then press the 'Verify' button to complete the challenge successfully. This sequence ensures that only the correct tile is selected, adhering to the CAPTCHA requirements.

## B.2. Text-like Challenge

**👤 User**

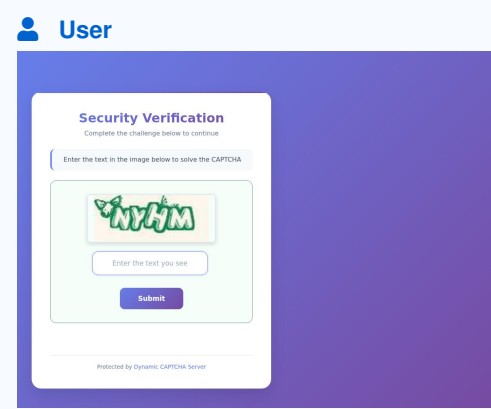

Solve the CAPTCHA as per the given instructions. You may need to interact with various elements such as checkboxes, image grids, sliders, or text inputs to complete the CAPTCHA challenge.

**🤖 Assistant**

I observe a text CAPTCHA on a security verification page with a warped image displaying 'WYHM'. The layout includes an input field and a submit button. I infer that the task requires typing the characters from the image into the input field. My plan is to focus on the input field, type 'WYHM' exactly as seen, and then press the submit button. This sequence follows the planned actions provided, ensuring the CAPTCHA is solved correctly.

**👤 User**

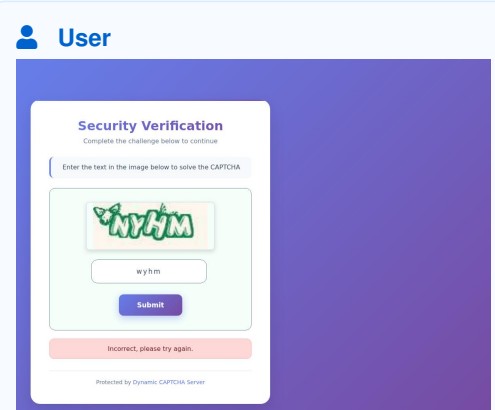

Continue solving the CAPTCHA. Observe the current state and take the next actions.

**🤖 Assistant**

I observe that the CAPTCHA scene involves a text entry challenge with the warped letters 'NYHM'. The layout includes an input field and a submit button, essential for completing the task. In my previous attempt, I misinterpreted the first letter as 'W' instead of 'N', leading to an incorrect submission of 'WYHM'. This mistake highlights the importance of accurately recognizing each character, especially when they are warped or stylized. The correct actions involve clicking on the input field at the precise location, typing 'NYHM' accurately, and then clicking the submit button. I plan to first focus on the input field at the correct relative point, ensuring I type the characters exactly as they appear, paying close attention to the letter 'N'. After typing 'NYHM', I will click the submit button at the specified location to complete the CAPTCHA successfully.

# C. Dynamic CAPTCHA System Design

This section provides a detailed description of the individual CAPTCHA challenge types in our dynamic CAPTCHA system. As introduced in Section 3.1, each challenge is designed to isolate and exercise a specific subset of fundamental interaction primitives required for modern CAPTCHA solving. While the main text focuses on the system-level design principles, this section documents the task formulations, visual layouts, and interaction requirements of each challenge variant.

- **Text-like Challenge** Figure 4 demonstrates text-like CAPTCHA challenges. These challenges present distorted alphanumeric strings against diverse backgrounds. Solving them requires the agent's ability to perform precise OCR and text entry. We utilize a dataset of distorted text images (hammer888, 2024) that are rendered within dynamically sized containers. The "Compact" variant imposes constraints on the viewpoint size, compelling the agent to adapt to the dense UI layouts commonly found on many webpages.

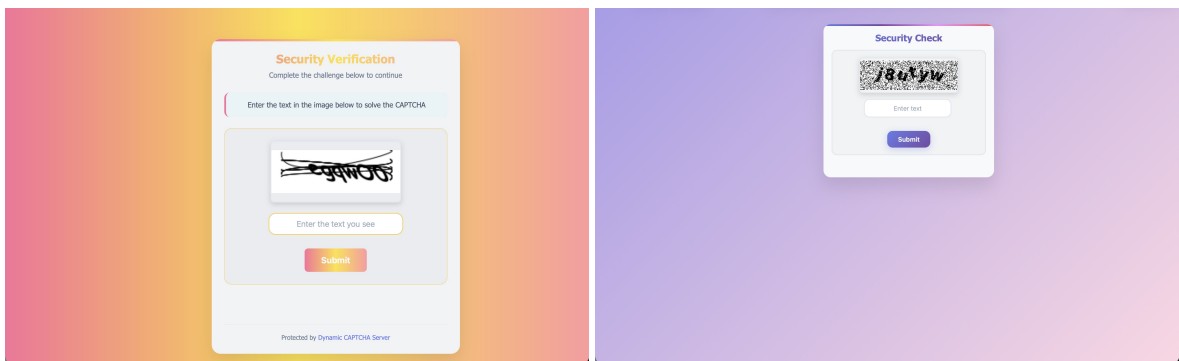

*(a)* Regular Text CAPTCHA Challenge      *(b)* Compact Text CAPTCHA Challenge

*Figure 4.* Text-like CAPTCHA Challenges

- **Grounding-related Challenge** Figure 5 shows a grounding-related CAPTCHA challenge. To train object recognition and precise coordinate clicking, we implement an Icon Selection challenge. The system renders a grid of Font Awesome icons with randomized rotations, colors, and positions. The agent must semantically map a textual instruction to a visual entity and execute a click event.

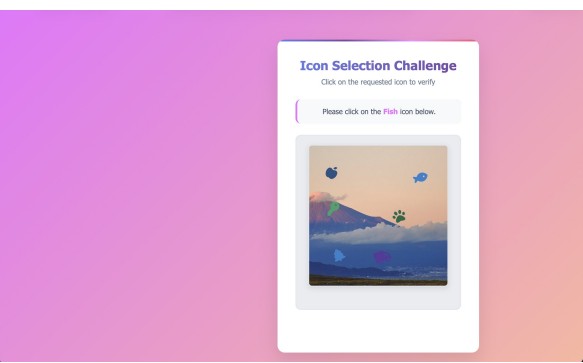

*Figure 5.* Icon Selection CAPTCHA Challenge

- **Drag-related Challenge** Figure 6 demonstrates drag-related CAPTCHA challenges. GUI agents often face challenges in performing continuous manipulation tasks. The *Slider* challenge simulates a puzzle-piece alignment task using vector-based masking. The agent's objective is to calculate the visual offset of a missing puzzle piece and drag a slider handle to align it accurately. The *Icon Match* challenge requires the agent to identify a pair of identical icons among distractors and drag one onto the other. This task trains the agent's object association and drag mechanics. Together, these two tasks effectively enforce visual alignment reasoning and fine-grained continuous control.

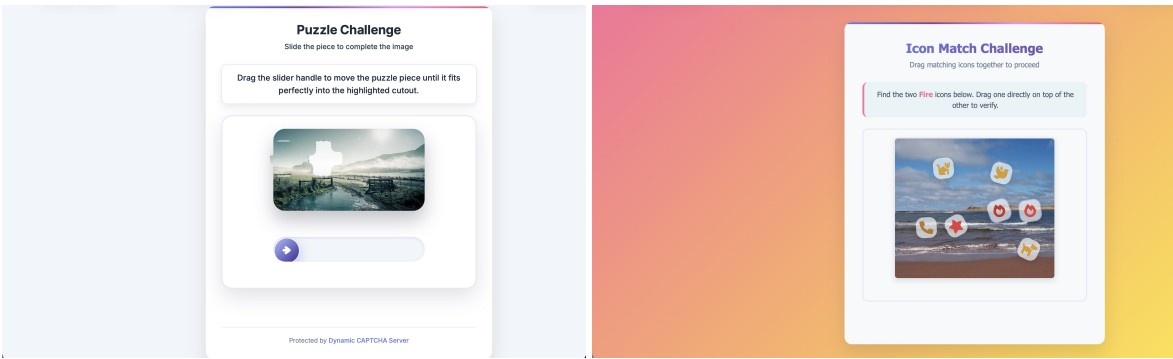

*(a)* Slider CAPTCHA Challenge          *(b)* Icon Match CAPTCHA Challenge

*Figure 6.* Drag-related CAPTCHA Challenges

- **Paged Challenge** Figure 7 shows paged CAPTCHA challenges. In this challenge, the target page is hidden within a paginated view, requiring the agent to navigate through multiple pages to locate and select the correct icon or image. This trains short-term memory and sequential planning.

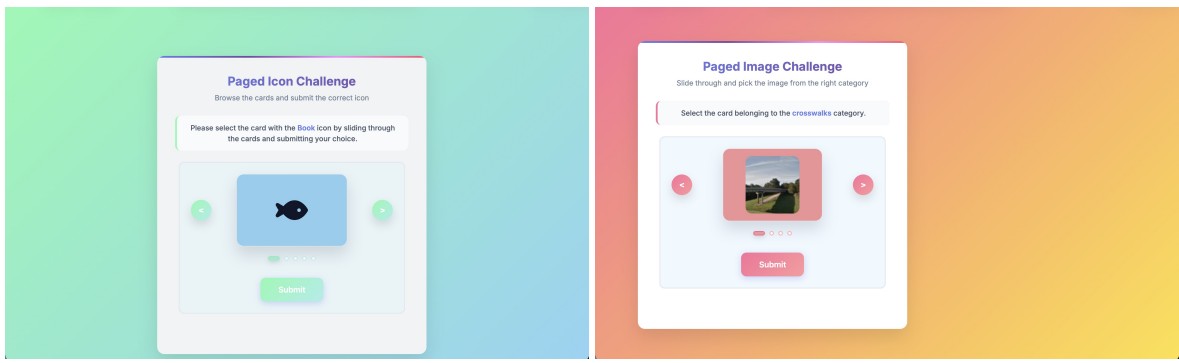

*(a)* Icon-based Paged CAPTCHA Challenge          *(b)* Image-based Paged CAPTCHA Challenge

*Figure 7.* Paged CAPTCHA Challenges

- **Image Grid Challenge** Figure 8 shows an image grid CAPTCHA challenge. Modeled after Google's reCAPTCHA v2, this challenge presents a $3 \times 3$ grid of images. The agent must select all tiles matching a specific semantic category. This specifically targets the model's ability to perform open-set visual classification and comprehensive skills to handle multi-step interactions.

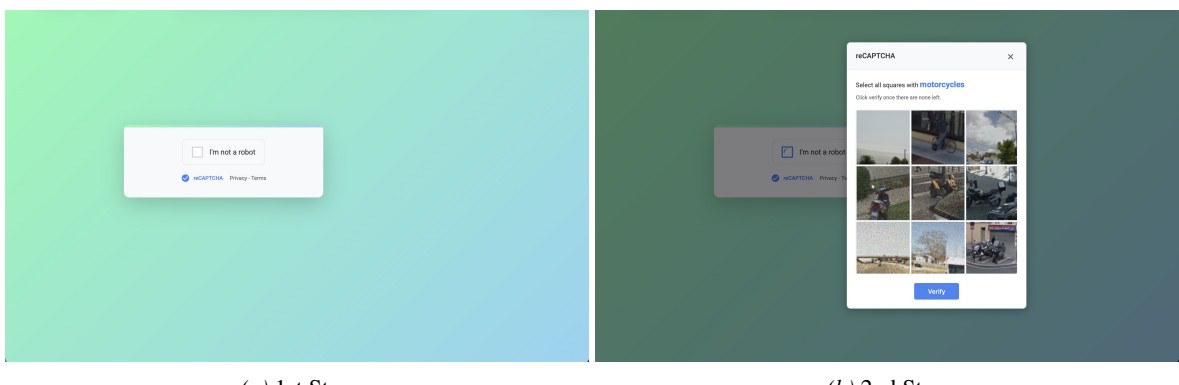

*(a)* 1st Step          *(b)* 2nd Step

*Figure 8.* Image Grid CAPTCHA Challenge

# D. More Details in Data Creation & Solution Evaluation Pipelines

In this section, we delve deeper into our data generation and curation pipeline, as well as the evaluation benchmark for CAPTCHAs and general GUI tasks.

## D.1. Prompt for Data Generation & Curation Pipeline

---

**Solution Trace Reasoning Prompt**

```
"You are documenting the internal reasoning for a CAPTCHA-solving assistant before
    it acts. \n"

f"Challenge type: {challenge_type}. \n"
f"Challenge details: {describe_challenge(challenge_type, solution_data)} \n"
f"Planned actions: {describe_actions(challenge_type, solution_data)}\n"
f"You are also provided with initial CAPTCHA screenshot and the annotated version
    of the screenshot with the actions steps the assistant should take. While you
    can use the annotations to understand the scene, you should not mention the
    existence of the annotations in your reasoning.\n"

"""
**REQUIREMENTS**
- You should think base on the provided images.
- Describe what you **observe** in the CAPTCHA scene (layout, visual cues, objects,
    etc.).
- Explain what you **infer** from those observations (what the task requires).
- Describe what you **plan** to do (actions) step by step. Make sure you exactly
    follow the order included in planned actions.
- Example output:
    "I observe that I am currently in a webpage that ask me solve a CAPTCHA. The
    CAPTCHA asks me to select the icon "book". Below the text instruction of the
    CAPTCHA, I can see a canvas with several icons of different color in the river
    background. On top left of the canvas, there's a purple icon that looks like
    "duck", to the right there's an icon that looks like "car", ...... On the
    second row I see a blue "book" icon, which may be the icon I should click .....
    I need to perform multiple actions to solve this CAPTCHA: I should click on the
    "book" icon on the second row to finish the task."

**CAPTCHA-specific Hints**
- For CAPTCHAs that use the "type" operation, remember to click on the input box
    first before typing in the answer.
- For Image Grid, you should describe each image block and make a judgment on
    whether the required element exists, then click on the correct image blocks.
- When a submit/verify button is present, end the plan by clicking it. Some slider
    and icon CAPTCHAs do not have a submit button and generally submit
    automatically after the main action is completed.
"""
```

---

**Correction Trace Reasoning Prompt**

```
"You are documenting the correction internal reasoning for a CAPTCHA-solving
    assistant after a failed attempt. \n"

"You are given the images of the CAPTCHA before and after your failed attempt, as
    well as an annotated image with action steps to solve the CAPTCHA (3 images in
    total). Although you have access to the ground truth actions and annotated
    image, you should pretend to find out the error by yourself from the failed
    attempt reasoning and updated images. You should never mention the ground truth
    actions or annotated image in your response."

f"Challenge type: {challenge_type}."
f"Challenge details: {describe_challenge(challenge_type, solution_data)}"
```

---

```
f"Previous reasoning: {model_reasoning}"
f"Correct actions: {solver_actions_formatted}"

f"""
**REQUIREMENTS**
- Your output should ONLY contain the thinking part, NOT the action part. The
    actions will be added separately.
- Describe what you **observe and infer** in the CAPTCHA scene (layout, visual
    cues, objects, etc.).
- Analyze what you **did wrong** in your previous attempt and why the correct
    actions are right.
- Describe what you **plan** to do (correct actions) step by step. Make sure you
    exactly follow the order included in correct actions.
- Example output:
    "I observe that I am currently in a webpage that asks me to solve a CAPTCHA.
    The CAPTCHA asks me to select the icon "book". Below the text instruction of
    the CAPTCHA, I can see a canvas with several icons of different color in the
    river background. On top left of the canvas, there's a purple icon that looks
    like "duck", to the right there's an icon that looks like "car", ...... On the
    second row I see a blue "book" icon, which may be the icon I should click. In
    my previous attempt, I incorrectly clicked on the "car" icon because I
    misidentified it as a "book". Looking at the second image, I can see that this
    action was wrong and the CAPTCHA failed. The correct approach is to click on
    the blue "book" icon on the second row, which clearly represents a book, not a
    car. I need to perform multiple actions to solve this CAPTCHA: I should click
    on the "book" icon on the second row to finish the task."
```

## D.2. Prompt for Dynamic CAPTCHA System & Halligan Benchmark

**System Prompt**

```
# Tools

You may call one or more functions to assist with the user query.

You are provided with function signatures within <tools></tools> XML tags:

{tool_desc}

For each function call, return a json object with function name and arguments
    within <tool_call></tool_call> XML tags:
<tool_call>
{{"name": <function-name>, "arguments": <args-json-object>}}
</tool_call>

# Response format

Response format for every step:
1) A single <tool_call>...</tool_call> block containing only the JSON: {{"name":
    <function-name>, "arguments": <args-json-object>}}.

Rules:
- Be brief: output concise thoughts.
- Do not output anything else outside those parts.
- If finishing, use action=terminate in the tool call.
```

**Initial User Prompt**

```
Solve the CAPTCHA as per the given instructions. You may need to interact with
    various elements such as checkboxes, image grids, sliders, or text inputs to
```

```
        complete the CAPTCHA challenge.
```

---

**Follow-up User Prompt**

```
Continue solving the CAPTCHA. Observe the current state and take the next actions.
```

## D.3. More Details in General GUI Agent Benchmarks

In this subsection, we provide additional information about the general GUI agent benchmark used in Section 4.4.

### D.3.1. ANDROID CONTROL

Android Control is a large-scale dataset comprising 15,283 demonstrations of human tasks performed in Android apps (Li et al., 2024). For every task, Android Control provides both high-level and low-level human-generated instructions describing it. In the high-level setting, the model is required to autonomously plan and execute actions over multiple steps, whereas in the low-level setting, the model follows human-annotated instructions to perform the next-step action.

Following the setting of Li et al. (2024), we randomly sample 500 step-action instances from the full AndroidControl test set to construct our evaluation subset. For grounding performance evaluation, we use UGround's pre-generated GPT-4o planning data for a more comparable element accuracy outcome (Gou et al., 2025). We report step accuracy under both high-level and low-level task settings. During inference, we set the temperature to 0 to reduce sampling variability and ensure stable model outputs.

---

**User Prompt**

```
Your task is to help the user identify the precise coordinates (x, y) of a specific
    area/element/object on the screen based on a description.

- Your response should aim to point to the center or a representative point within
    the described area/element/object as accurately as possible.
- If the description is unclear or ambiguous, infer the most relevant area or
    element based on its likely context or purpose.
- Your answer should be a single string (x, y) corresponding to the point of the
    interest.

Description: {description}

Answer:"""
```

---

### D.3.2. SCREENSPOT-V2

ScreenSpot-V2 (Wu et al., 2024) is a large-scale benchmark for evaluating visual grounding and spatial localization in GUI environments. The benchmark challenges models to identify precise screen coordinates corresponding to a natural-language description of a target UI element, requiring fine-grained visual perception and accurate reasoning over layout, affordances, and contextual cues. ScreenSpot-V2 contains 1,273 tasks across web, desktop, and mobile domains. It also addresses approximately 11.32% of samples in the original ScreenSpot dataset (Cheng et al., 2024) that contained errors or ambiguities.

We adapt the standard setting without a planner, directly using the original instructions from ScreenSpot. We report grounding accuracy by computing if the predicted location falls within the ground truth element's bounding box. During inference, we set the temperature to 0 to reduce sampling variability and ensure stable model outputs. For the 32B ReCAP variants, we tuned the prompt by removing the system prompt and adding explicit instructions in the user prompt. For all other models, we use the original ScreenSpot instructions as the user prompts without modification.

---

**System Prompt**

```
You are a helpful assistant. The user will give you an instruction, and you MUST
    left click on the corresponding UI element via tool call. If you are not sure
    about where to click, guess a most likely one.\n\n# Tools
```

---

```
You may call one or more functions to assist with the user query.

You are provided with function signatures within <tools></tools> XML tags:
<tools>
{"type": "function", "function": {"name": "computer_use", "description": "Use a
    mouse to interact with a computer.\n* The screen's resolution is 1000x1000.\n*
    Make sure to click any buttons, links, icons, etc with the cursor tip in the
    center of the element. \n* You can only use the left_click action to interact
    with the computer.", "parameters": {"properties": {"action": {"description":
    "The action to perform. The available actions are:\n* 'left_click': Click the
    left mouse button with coordinate (x, y).", "enum": ["left_click"], "type":
    "string"}, "coordinate": {"description": "(x, y): The x (pixels from the left
    edge) and y (pixels from the top edge) coordinates to move the mouse to.
    Required only by 'action=left_click'.", "type": "array"}, "required":
    ["action"], "type": "object"}}}
</tools>

For each function call, return a json object with function name and arguments
    within <tool_call></tool_call> XML tags:
<tool_call>
{"name": <function-name>, "arguments": <args-json-object>}
</tool_call>
```

**User Prompt for 32B ReCAP Variants**

```
Output the bounding box in the image corresponding to the instruction
    "{instruction}" with grounding. Your output should be in the format of a JSON
    array inside ```json``` code block, like this:
```json
[{{"bbox_2d": [x1, y1, x2, y2]}}]
```.
Make sure to strictly follow this format.
```

### D.3.3. MULTIMODAL-MIND2WEB

Multimodal-Mind2Web is the multimodal version of Mind2Web, a dataset built for developing and evaluating generalist agents for the web that can follow language instructions to complete complex tasks on any website. We evaluate baseline model and ReCAP models' Web grounding performance using Multimodal-Mind2Web (MM-Mind2Web) (Zheng et al., 2024b). MM-Mind2Web is the multimodal version of Mind2Web (Deng et al., 2023), which consists of three parts in its test dataset, a total of 1013 tasks. We base our evaluation pipeline on UGround's offline evaluation pipeline (Gou et al., 2025). As we evaluate our model for grounding performance, we use UGround's pre-generated GPT-4o planning data for a more comparable element accuracy outcome. UGround's planned actions dataset consists in total of 4751 actions for the three parts of MM-Mind2Web dataset. Given the way we hosted our ReCAP model, we need to supply the system prompt through the hosting API. We intentionally input system prompt and user prompt in each API query to the Qwen3-VL baseline as well as the ReCAP models. For OpenAI CUA, it does not require a system prompt. We select "`browser`" environment for "`computer_use_preview`" tool instead.

**System Prompt**

```
Your task is to help the user identify the precise coordinates (x, y) of a specific
    area/element/object on the screen based on a description.

- Your response should aim to point to the center or a representative point within
    the described area/element/object as accurately as possible.
- If the description is unclear or ambiguous, infer the most relevant area or
    element based on its likely context or purpose.
- Your answer should be a single string (x, y) corresponding to the point of the
    interest.
```

---

**User Prompt**

```
Your task is to help the user identify the precise coordinates (x, y) of a specific
    area/element/object on the screen based on a description.

- Your response should aim to point to the center or a representative point within
    the described area/element/object as accurately as possible.
- If the description is unclear or ambiguous, infer the most relevant area or
    element based on its likely context or purpose.
- Your answer should be a single string (x, y) corresponding to the point of the
    interest.

Description: {description}

Answer:
```

---

**User Prompt for OpenAI CUA**

```
Your task is to help the user identify the precise coordinates (x, y) of a specific
    area/element/object on the screen based on a description.

- Your response should aim to point to the center or a representative point within
    the described area/element/object as accurately as possible.
- If the description is unclear or ambiguous, infer the most relevant area or
    element based on its likely context or purpose.
- You must left click on the center of corresponding point of interest regardless
    type of element. Do not drag any element.

Description: {description}
```

### D.3.4. UI-VISION

UI-Vision (Nayak et al., 2025) is a desktop-centric GUI benchmark that evaluates visual perception and interaction through element-grounding and layout-grounding tasks. We use it as a harder desktop-focused complement to Android Control, ScreenSpot-V2, and Multimodal-Mind2Web. UI-Vision does not natively use Qwen3-style bounding-box outputs, so we use a custom parser for model outputs and disable thinking for all evaluated models. In Table 3, we report UI-Vision's overall element-grounding accuracy as the Element row result and mean IoU as the Layout row result.

---

**User Prompt for Element Grounding Task**

```
Output the bounding box in the image of the UI element corresponding to the
    instruction "{instruction}" with grounding. The coordinates should be relative
    ranging from 0 to 1000, relative to the actual image width and height (i.e. all
    x and y values are in [0, 1000]). Report bbox coordinates in JSON format as
    {"bbox_2d": [x1, y1, x2, y2]}. Return exactly one JSON object and nothing else.
```

---

**User Prompt for Layout Evaluation Task**

```
A functional region is a specific part of the UI that groups tools or elements
    serving a particular purpose. Your task is to identify the bounding box for the
    following functional region. Name: {name}

Explanation: {explanation}

The coordinates should be relative ranging from 0 to 1000, relative to the actual
    image width and height (i.e. all x and y values are in [0, 1000]). Report bbox
    coordinates in JSON format as {"bbox_2d": [x1, y1, x2, y2]}. Return exactly one
    JSON object and nothing else.
```

# E. Additional Details and Analyses

## E.1. Executable Action Space and Ground-truth Actions

ReCAP follows the Qwen3-VL computer-use action format and extends it to allow multiple executable actions in one model response. Each atomic action can be abstracted as a tuple (`action_type`, `value`, `point_2d`), where `point_2d` is expressed in a 1000×1000 relative coordinate grid. The executor maps these relative coordinates to the rendered browser viewport before dispatching Playwright actions. The supported atomic actions are:

*Table 6.* Unified action space. Each action is a tuple (`action_type`, `value`, `point_2d`).

| action_type | value | point_2d | details |
|---|---|---|---|
| Left click | None | [x,y] | Execute Qwen3-VL action `left_click`; click the left mouse button at the target coordinate. |
| Right click | None | [x,y] | Execute Qwen3-VL action `right_click`; click the right mouse button at the target coordinate. |
| Middle click | None | [x,y] | Execute Qwen3-VL action `middle_click`; click the middle mouse button at the target coordinate. |
| Double click | None | [x,y] | Execute Qwen3-VL action `double_click`; double-click the left mouse button at the target coordinate. |
| Type | input text | [−100,−100] | Execute Qwen3-VL action `type`; fill a detected input field when possible, otherwise type at the current focus. |
| Keyboard press | key name or key sequence | [−100,−100] | Execute Qwen3-VL action `key`; press a single key or a key combination. |
| Scroll | pixel amount | [−100,−100] | Execute Qwen3-VL action `scroll`; scroll by the specified signed pixel amount. |
| Mouse move | None | [x,y] | Execute Qwen3-VL action `mouse_move`; move the cursor to an initial coordinate, typically before a drag. |
| Left click drag | optional duration | [x,y] | Execute Qwen3-VL action `left_click_drag`; hold the left button from the current cursor position and drag to the target coordinate. |
| Wait | seconds | [−100,−100] | Execute Qwen3-VL action `wait`; pause execution for a specified duration to allow UI updates. |
| Terminate | success/failure status | [−100,−100] | Execute Qwen3-VL action `terminate`; signal the end of the current task. |

In the actual model output, each action is emitted as a Qwen-style `computer_use` tool call. For example, a click at the relative coordinate [420,615] is represented as:

```
<tool_call>
{"name":"computer_use","arguments":{"action":"left_click","coordinate":[420,615]}}
</tool_call>
```

Multiple actions can be emitted in one response by listing several `computer_use` calls inside the same `<tool_call>` block. For example, a text CAPTCHA can be solved by clicking the input field, typing the answer, and clicking submit:

```
<tool_call>
{"name":"computer_use","arguments":{"action":"left_click","coordinate":[210,780]}},
{"name":"computer_use","arguments":{"action":"type","text":"a7kx9"}},
{"name":"computer_use","arguments":{"action":"left_click","coordinate":[820,780]}}
</tool_call>
```

Ground-truth actions are generated from the API of each CAPTCHA instance. Text-like challenges click the input box, type the ground-truth string, and submit. Icon Selection clicks the target icon center. Slider moves to the handle and drags to the target offset. Icon Match moves to the source icon and drags it onto the matching target. Paged challenges issue page-switching actions until the target page is visible, then click the target. Image Grid challenges click all matching tiles and then verify. For multi-action cases, actions are emitted as an ordered list of `computer_use`-compatible tuple actions. The parser validates action names and required parameters and recovers from minor formatting inconsistencies; failed environment feedback can then be used to construct self-correction traces.

**E.2. Additional Real-world CAPTCHA Evaluation Results**

In this section, we provide a full real-world CAPTCHA comparison on the Halligan benchmark in Table 7, including Qwen3-VL Thinking and Instruct baselines, all four finetuned ReCAP variants, GPT-5.4, and Gemini-3-Flash-Preview. For the Instruct variants, ReCAP-8B (Instruct) and ReCAP-32B (Instruct) improve over their corresponding Qwen3-VL baselines by 14.50 pp and 11.42 pp on average, respectively, and both outperform Halligan by a small margin on average. The Thinking variants show a similar baseline-improvement trend: ReCAP-32B (Thinking) improves over Qwen3-VL-32B-Thinking by 12.58 pp on average and outperforms its baseline on 25 of 26 CAPTCHA types. It also achieves best or second-best results among the non-frontier models on many interaction-intensive challenges, including *recaptchav2*, *hcaptcha*, and multiple *Arkose Lab* variants, where precise spatial reasoning and continuous control are critical.

The frontier-model results show that proprietary agents remain highly competitive. Gemini-3-Flash-Preview obtains the highest average score in the full comparison, while GPT-5.4 is close to the ReCAP Instruct variants overall. However, ReCAP remains stronger on several categories, including multiple interaction-heavy Arkose and GeeTest variants. These mixed results reinforce that our real-world evaluation should be interpreted as zero-shot transfer from dynamic training rather than uniform state-of-the-art performance.

*Table 7.* **Full real-world CAPTCHA comparison including Qwen3-VL baselines, all finetuned ReCAP variants, Halligan, and frontier models.** Our models are evaluated in a *zero-shot* setting without training on this real-world CAPTCHA dataset, whereas Halligan (H) is *explicitly designed* for these CAPTCHA types. T and I denote Thinking and Instruct variants, respectively. All numbers are solve rates (%).

| CAPTCHA Type | ReCAP -8B (T) | ReCAP -8B (I) | ReCAP -32B (T) | ReCAP -32B (I) | H | Qwen -8B (T) | Qwen -8B (I) | Qwen -32B (T) | Qwen -32B (I) | GPT-5.4 | Gemini 3-Flash |
|---|---|---|---|---|---|---|---|---|---|---|---|
| tencent/vtt | 32 | 24 | 41 | 39 | 8 | 29 | 42 | 28 | 37 | 48 | 77 |
| mtcaptcha | 21 | 65 | 22 | 73 | 66 | 14 | 63 | 10 | 56 | 42 | 58 |
| yandex/text | 36 | 74 | 49 | 85 | 82 | 27 | 81 | 24 | 80 | 69 | 68 |
| botdetect | 54 | 83 | 61 | 72 | 80 | 52 | 74 | 52 | 78 | 50 | 68 |
| recaptchav2 | 27 | 68 | 63 | 74 | 61 | 30 | 12 | 23 | 36 | 77 | 9 |
| baidu/rotate | 14 | 21 | 33 | 22 | 79 | 5 | 0 | 6 | 0 | 16 | 31 |
| hcaptcha | 21 | 47 | 26 | 63 | 2 | 24 | 37 | 20 | 62 | 70 | 67 |
| geetest/slide | 36 | 68 | 26 | 80 | 16 | 9 | 0 | 7 | 16 | 38 | 41 |
| lemin | 1 | 18 | 8 | 12 | 0 | 0 | 0 | 0 | 0 | 4 | 25 |
| amazon/waf | 16 | 20 | 10 | 27 | 6 | 4 | 15 | 4 | 23 | 56 | 53 |
| funcaptcha/counting | 41 | 29 | 42 | 36 | 54 | 40 | 32 | 41 | 33 | 36 | 68 |
| funcaptcha/hand_number | 34 | 34 | 51 | 26 | 40 | 21 | 32 | 31 | 34 | 19 | 36 |
| funcaptcha/galaxies | 98 | 99 | 97 | 100 | 100 | 96 | 100 | 95 | 99 | 100 | 99 |
| funcaptcha/dice_pair | 39 | 30 | 43 | 35 | 27 | 32 | 33 | 36 | 20 | 36 | 56 |
| funcaptcha/card | 40 | 33 | 41 | 24 | 25 | 28 | 38 | 25 | 31 | 36 | 50 |
| funcaptcha/square_icon | 48 | 46 | 40 | 54 | 82 | 43 | 63 | 37 | 73 | 78 | 76 |
| funcaptcha/rotated | 46 | 82 | 57 | 85 | 85 | 56 | 79 | 62 | 74 | 77 | 94 |
| arkose/3d_rollball | 8 | 50 | 32 | 38 | 1 | 5 | 0 | 1 | 3 | 27 | 14 |
| arkose/dice_match | 7 | 51 | 15 | 37 | 8 | 6 | 0 | 9 | 0 | 23 | 31 |
| arkose/orbit_match | 17 | 38 | 20 | 30 | 8 | 1 | 0 | 1 | 7 | 36 | 33 |
| arkose/rockstack | 18 | 57 | 22 | 33 | 13 | 11 | 4 | 6 | 14 | 24 | 28 |
| arkose/numbermatch | 10 | 43 | 28 | 39 | 8 | 8 | 3 | 9 | 9 | 34 | 28 |
| geetest/icon | 5 | 15 | 10 | 15 | 2 | 3 | 6 | 1 | 19 | 68 | 30 |
| geetest/iconcrush | 13 | 6 | 14 | 8 | 70 | 9 | 14 | 6 | 5 | 1 | 33 |
| geetest/gobang | 0 | 0 | 3 | 0 | 36 | 0 | 4 | 2 | 1 | 0 | 22 |
| yandex/kaleidoscope | 12 | 8 | 11 | 3 | 47 | 4 | 0 | 2 | 3 | 7 | 9 |

### E.3. Training Setup

We train all ReCAP variants with full-parameter supervised finetuning. Table 8 summarizes representative training settings; model path and output directory are changed for each base model variant.

*Table 8.* Training setup for ReCAP models.

| Configuration | Value |
|---|---|
| Hardware | $4\times$ NVIDIA B200 GPUs |
| Base model | Qwen3-VL 8B/32B Thinking and Instruct variants |
| Finetuning method | Full-parameter SFT with DeepSpeed ZeRO-2 |
| Precision | bf16 |
| Maximum image pixels | 5,720,064 |
| Cutoff length | 12,288 tokens |
| Per-device batch size | 4 |
| Gradient accumulation | 8 steps |
| Gradient checkpointing | Enabled |
| Learning rate | $1 \times 10^{-5}$ |
| Scheduler and warmup | Cosine decay with 5% warmup |
| Optimizer | AdamW fused |
| Epochs | 1 |
| Validation split | 1% |
| Loss weights | $\lambda_{\text{think}} = 0.5, \lambda_{\text{act}} = 0.5$ |

### E.4. Data Mixture and Capacity

In this section, we analyze how the training data mixture and model capacity affect CAPTCHA-solving performance and general GUI capability retention for the Thinking model variants. We mix general GUI trajectories into training to preserve broad interaction ability while adding CAPTCHA-specific supervision. A data-scale sweep on ReCAP-8B (Thinking) shows that increasing CAPTCHA data improves performance up to the final scale used in the paper, after which gains saturate.

*Table 9.* CAPTCHA data-scale sweep for ReCAP-8B (Thinking) used to choose the final training mixture.

| Model | CAPTCHA Data Scale | Overall SR (%) | Avg. Steps |
|---|---|---|---|
| ReCAP-8B (Thinking) | 50% | 62.90 | 1.60 |
| ReCAP-8B (Thinking) | 100% | 71.90 | 1.54 |
| ReCAP-8B (Thinking) | 150% | 72.20 | 1.52 |

Model capacity also shapes how well CAPTCHA specialization transfers to general GUI tasks. Among the Thinking variants, ReCAP-8B degrades on several general GUI benchmarks, and an additional experiment with Qwen3-VL-4B-Thinking under the same setting shows an even larger relative drop on ScreenSpot-V2 (21.15% versus its baseline). In contrast, ReCAP-32B (Thinking) better absorbs CAPTCHA-specific supervision while retaining general GUI behavior.

### E.5. Failure Modes

In this section, we summarize the main failure modes observed from manual inspection of failed held-out dynamic CAPTCHA cases. First, highly distorted OCR remains difficult, especially confusions such as "1" versus "I" and "0" versus "O". Second, multi-step tasks can suffer from error accumulation: an early wrong click, drag, or page transition changes the state and makes later correction harder. Third, visual-semantic recognition errors occur when the model misidentifies icons or image-grid objects. These failures indicate that the main remaining bottlenecks are perception robustness and long-horizon recovery.

