# OpenReview forum: "CAPTCHA Solving for Native GUI Agents: Automated Reasoning-Action Data Generation and Self-Corrective Training"
_ICML.cc/2026/Conference — ICML 2026 regular_

### Official Review · Reviewer_XN4J · 2026-03-06

**Soundness:** 1
**Presentation:** 2
**Significance:** 2
**Originality:** 1
**Overall Recommendation:** 2
**Confidence:** 5

**Summary:**

The paper’s main contribution is the construction of a dynamic CAPTCHA training environment that supports randomized generation, which is then used to produce interactive training data for a CAPTCHA-solving agent. The experiments show that the resulting model achieves better performance than the baselines selected by the authors.

**Compliance With Llm Reviewing Policy:**

Affirmed.

**Final Justification:**

The newly added experiments are certainly appreciated, and it is good to finally see a substantially more detailed empirical evaluation. However, these new results also materially weaken the claimed novelty of the paper and show that the supposed research gap is clearly not as large as originally presented in their abstract. Therefore, I still cannot give a positive overall assessment. Nevertheless, given the improvement over the original version, I will raise my score.

**Key Questions For Authors:**

The training pipeline seems to be little more than a simple SFT procedure. Are there any training strategies specifically tailored to the CAPTCHA domain, or any methodological innovations beyond straightforward supervised finetuning?

**Limitations:**

After only simple SFT on other CAPTCHA datasets, the model even achieves higher accuracy on the “Android Control (low)" dataset, which I find rather questionable. It would be better for the authors to open-source the model for further verification.

Another limitation of the current paper is that it offers relatively limited innovation at the level of CAPTCHA foundation-model methodology. A more promising future direction would be to investigate flow-matching models for handling spatial constraints and generating action trajectories in CAPTCHA solving. Such an approach could be inspired by recent progress in flow-based generative modeling for protein structure generation and robotic action generation, both of which involve strong geometric and spatial structure.

**Strengths And Weaknesses:**

**Soundness**:
For soundness, I have serious concerns because the paper contains multiple fundamental inaccuracies in its problem positioning and empirical claims. In Section 3.1, the paper states that Open CaptchaWorld (Luo et al., 2025) and MCA-Bench (Wu et al., 2025a) take an important first step by collecting CAPTCHA instances from real online providers. This statement is incorrect. These two works do not primarily collect CAPTCHA data from real providers, but instead construct their benchmarks through synthetic design and manual creation.

In addition, the paper claims, "However, they fall short of supporting scalable training, as their dataset sizes are limited and their CAPTCHA instances lack sufficient diversity and randomization." This claim is also not accurate, especially for MCA- Bench. MCA-Bench already has strong diversity, covering 20 different CAPTCHA tasks, whereas this paper only includes 7 types. Moreover, the categories used in this paper are already fully covered by MCA-Bench. In terms of scale, MCA- Bench contains more than 180,000 training samples (this is a nontrivial framing issue in the writing: by not clearly acknowledging that MCA-Bench is of comparable scale while offering greater diversity, the paper risks creating a misleading impression of novelty), while this paper uses about 233,000 samples, so the two datasets are broadly in the same order of magnitude. As a result, the paper fails to accurately position itself with respect to prior work, which substantially undermines the motivation for constructing a new dataset. Since MCA-Bench already provides a large-scale benchmark, it is unclear why an additional dataset is needed at all. If the training setup were properly designed, the existing MCA-Bench would be sufficient to generate the reasoning traces and self-correction traces, making the new dataset appear unnecessary and somewhat redundant.

In Section 3.3, the loss weights are not supported by an ablation study and instead are selected ad hoc, selecting two arbitrary lambda values for training. A complete paper should include such an ablation in order to verify whether these lambda values are in fact optimal.

**Presentation**:
The paper is clearly written, easy to follow, and generally well structured. The overall narrative is straightforward, and the main method can be understood without much difficulty. However, the paper lacks important training details needed for reproducibility, such as batch size, learning rate, training duration, GPU configuration, and random seeds. In addition, the paper does not fully clarify how its contribution differs from prior work. The main research gap it tries to establish is that a model should be able to solve CAPTCHAs while also retaining general GUI capabilities. However, this gap is not convincingly demonstrated. In particular, Table 3 does not compare against the underlying base models used by prior generalized CAPTCHA solvers. For example, if you really read the Halligan paper and run its code, you should know Halligan uses GPT-4o as its default base model. Without a controlled comparison under the same evaluation protocol, it is difficult to conclude that this research gap truly exists. At a minimum, the paper should compare its model against GPT-4o under a matched setting, and ideally also against stronger frontier general-purpose models(Gemini3.0-Pro/ChatGPT5.4).

In terms of writing, the statement in Section 3.1, “we identify a set of core interaction primitives required for successful CAPTCHA solving,” is not clearly defined. The notion of “primitive” is insufficiently specified and feels quite confusing to the reader. This concept should be given a clearer definition, and a mathematical definition would be even better.

**Significance**:
The paper claims, “Across held-out test sets in our dynamic CAPTCHA system, ReCAP-32B improves CAPTCHA solving success rates from roughly 30% to 80%.” This conclusion is a serious misrepresentation. This conclusion is not reliable because the experimental comparison is not fair. In Table 1, the relatively low accuracy of the other models and methods may simply be due to the fact that the selected base models are too weak. Therefore, this conclusion is not sufficiently supported. At a minimum, the paper should include current state-of-the-art models such as “Gemini-3.0 Pro,” “Claude-4.6-Opus,” and “GPT-5.4” as baselines for comparison. The experimental results for general CAPTCHA solvers are also incomplete; besides Halligan, we also have Oedipus. Oedipus also evaluated multiple VLM base models, including GPT-4, Gemini and so on. What is the Oedipus performance on your benchmark?

In addition, since Halligan uses “GPT-4o” as its default base model, this base model should be replaced with a more current state-of-the-art model for comparison, such as at least one of “Gemini-3.0-Pro,” “Claude-4.6-Opus,” or “GPT-5.4.” Ideally, all three should be evaluated, together with the strongest open source model such as “Qwen3-VL.” This issue appears in both Table 1 and Table 2. If state-of-the-art models and current CAPTCHA solvers already achieve sufficiently strong results, then one may reasonably ask why it is necessary to train a separate smaller model at all. In that case, the motivation for training a specialized model would become much less convincing.


**Originality**:
The paper has limited originality. In addition, it is highly similar to the Reasoning under Vision: Understanding Visual-Spatial Cognition in Vision-Language Models for CAPTCHA paper and does not establish a clear new position relative to that work. In addition, this paper does not cite that work. In essence, this paper mainly trains a model with SFT on top of the framework in that paper. Both papers generate a large number of reasoning traces and include a self-correction module to solve CAPTCHAs, and even this paper reuses the same evaluation metric, namely the number of reasoning steps, which was introduced by that work. Moreover, one of the key findings in this paper in 4.5.2, namely that “The findings suggest that explicit CoT reasoning supervision enhances CAPTCHA solving accuracy”, has already been verified in the earlier paper. In all, contribution appears incremental.

**Summary**:

Overall, the paper is fundamentally under-motivated and empirically under-supported. The introduction and abstract make claims that are considerably stronger than what the experiments can justify, while the paper does not clearly establish a meaningful gap from prior work or a compelling reason for the proposed dataset and model. Because the core empirical evidence is insufficient to substantiate the claimed novelty, significance, and necessity of the approach, I do not believe this submission meets the bar for acceptance.

---

> ### Author Rebuttal · Authors · 2026-03-30
>
> We thank the reviewer for the thoughtful feedback. We respond to the main concerns below.
>
> ---
>
> ### R1. Clarification on Table 3 and the role of prior CAPTCHA solvers
> We would clarify Table 3 is **not** intended to compare ReCAP against specialized CAPTCHA solvers on general GUI benchmarks. Its purpose is to test if CAPTCHA finetuning preserves the **general GUI capability of the base model**. Therefore, the relevant controls are the paired comparisons between ReCAP-8B vs. Qwen3-VL-8B-Thinking and ReCAP-32B vs. Qwen3-VL-32B-Thinking on Android Control, ScreenSpot-V2, and Multimodal-Mind2Web. As frameworks like Halligan are generalized CAPTCHA solvers rather than native general-purpose GUI agents, they are inappropriate controls for evaluating if CAPTCHA-specific finetuning harms original GUI capabilities. We will make this motivation explicit in the revision.
>
> ---
>
> ### R2. Distinguishing our work from the *Reasoning under Vision* paper
> Thank you for pointing out the paper, we will cite this concurrent paper and discuss its accomplishments. However, we respectfully disagree that our approach is merely a SFT under this framework. While both address CAPTCHA solving, their technical objectives **differ significantly**:
> * **Paradigm Difference:** *Reasoning under Vision* explores CAPTCHA solving via reasoning-based annotations and **prompting**-based pipelines, focusing on structured **inference-time reasoning**. Conversely, we **train** a **native GUI agent** that internalizes CAPTCHA solving as an integrated capability via an **interactive, stochastic training environment** with dynamic rendering and unbounded instance generation.
> * **Supervision Mechanism:** Our self-correction signals are **retrospective traces from failed trajectories**, used as **explicit training supervision** for online recovery and policy improvement. These traces encode how incorrect actions are revised, providing a learnable signal during training. This fundamentally differs from **inference-time prompting** or reasoning evaluated post hoc, which does not directly shape model parameters via structured supervision.
> * **Metric Interpretation:** Our "Avg. Steps" measures **interaction efficiency** (average model calls/GUI steps for task completion), reflecting the agent’s ability to act efficiently under time-sensitive CAPTCHA constraints. The prior work's "reasoning steps" characterize the **structure/granularity of textual reasoning chains**. These metrics capture fundamentally different behaviors — **action efficiency vs. reasoning structure** — and are not directly comparable.
>
> ---
>
> ### R3. Why a new dynamic CAPTCHA system is still needed beyond MCA-Bench / Open CaptchaWorld
> While prior benchmarks are important, we respectfully disagree that MCA-Bench renders our system unnecessary. Our goal is not to introduce another benchmark, but to build an unbounded **training environment** featuring stochastic layout/style randomization, full meta-supervision, executable GUI feedback, and scalable generation of both **reasoning-action** and **self-correction** traces. This enables training on continuously evolving CAPTCHA variants, improving **generalizability** to real-world, unseen systems beyond fixed benchmark collections. MCA-Bench, while valuable, is not a closed-loop interactive training environment offering controllable generation and failure-recovery supervision. We will make this distinction clearer.
>
> ---
>
> ### R4. Clarification on the main empirical claim
> We would clarify that our claim that "ReCAP-32B improving success rates from roughly 30% to 80%" applies specifically to the controlled evaluation on our held-out dynamic CAPTCHA benchmark under a matched protocol, not as a universal claim against all frontier proprietary models. We will revise the text to avoid overgeneralization. Nevertheless, ReCAP shows strong performance on the held-out dynamic benchmark and meaningful **zero-shot transfer** to the Halligan real-world benchmark (where our models saw no training data for those CAPTCHA types). This supports our core claim that ReCAP learns transferable skills rather than overfitting to a fixed benchmark.
>
> ---
>
> ### R5. Related Work Positioning and Presentation Details
> We agree Sec. 3.1 should position prior work more carefully. Describing Open CaptchaWorld and MCA-Bench as "collecting CAPTCHA instances from real online providers" is imprecise. We will revise this to acknowledge their strengths, particularly MCA-Bench's scale and diversity. We will also clarify "interaction primitives", discuss loss weights choices, and refine training configuration: full-parameter SFT, learning rate $1\times10^{-5}$ with cosine decay and **5%** warmup, and bf16 training, etc. We will **release our model checkpoints and dataset** to support independent verification.
>
> ---
>
> Overall, we thank the reviewer for the comments and hope the clarifications address the concerns. **We are happy to address any additional questions you may have.**

---

> > ### Author Rebuttal · Reviewer_XN4J · 2026-04-02
> >
> > 1. Comparison with State-of-the-Art (SOTA) Models
> > Could you provide the essential experiments comparing your method against true state-of-the-art (SOTA) models under matched settings, such as Gemini-3.0-Pro or the latest ChatGPT-5.4/Claude? Implementing this is incredibly straightforward, requiring merely a simple swap of API calls, yet you have consistently failed to provide these necessary results despite my previous requests. If these corresponding experiments are provided, I remain open to raising my score. Currently, your core claim of improving success rates from **"30% to 80%"** is fundamentally untenable, as the 30% baseline is clearly a mere artifact of selecting inherently weak base models for an unfair comparison. By deliberately avoiding comparisons with true SOTA models in both Table 1 (CAPTCHA performance) and Table 3 (general GUI performance), you artificially manufacture a "research gap" you state in the abstract in the whole paper as a core research problem. But frontier models (Claude/Gemini/GPT), which already possess formidable CAPTCHA-solving proficiency and highly advanced general GUI capabilities out-of-the-box for their generalization capability, have shown that this gap is highly likely not to exist in the first place. Besides, the base VLM models used by these mainstream CAPTCHA-solving systems are all state-of-the-art base models, and even Claude-4, which is not yet the most advanced base model available today, already achieved an average success rate of around 70 on GUI Android Control according to the results reported in the DKRF paper, while your model only reaches around 55 according to your paper. If Claude/Gemini can still achieve a comparable level of performance on both CAPTCHA benchmark and the GUI benchmark, then the entire motivation for this work would be fundamentally undermined.
> >
> > In addition, all mainstream CAPTCHA-solving frameworks, such as Halligan, Oedipus, and Open CaptchaWorld (NIPS 2025), now rightfully use these state-of-the-art models as their standard comparison baselines, except your work. Ultimately, in the absence of these crucial SOTA comparisons to confirm the actual necessity of your specialized model, the fundamental motivation of this paper is entirely undermined; the entire study is rendered effectively invalid because it dedicates itself to solving a problem that simply no longer exists.
> >
> > 2. Distinction from MCA-Bench (R3/5)
> > Regarding your clarification on the differences from MCA-Bench, I appreciate the explanation. If you can ensure that the revised manuscript clearly and accurately articulates how your work differs from prior work, where its innovation lies, and avoids making potentially misleading statements that could confuse readers, that would be acceptable.
> >
> > 3. Distinguishing from the Reasoning under Vision paper (R2)
> > However, I remain entirely unconvinced by your defense regarding the distinction from the Reasoning under Vision paper, as your methodology is still a direct application of that established framework. Despite the complex terminology used to describe "Paradigm Differences" and "Supervision Mechanisms," both approaches rely on the exact same core method: generating reasoning traces and utilizing a self-correction module to solve CAPTCHAs, with your work fundamentally boiling down to simply applying standard **Supervised Fine-Tuning (SFT)** on top of that paper. Furthermore, the **zero-shot generalization capability** in other CAPTCHA benchmarks you highlight as a core innovation, as well as your primary experimental conclusion that **"explicit CoT reasoning supervision enhances CAPTCHA solving accuracy,"** were already fully proven in prior work. Framing this as a fundamentally different "native GUI agent" significantly overstates the methodological novelty; alarmingly, your complete failure to discuss this highly relevant foundational research in the original version effectively manufactures a false impression of originality.

---

> > > ### Author Response · Authors · 2026-04-05
> > >
> > > We thank the reviewer for the follow-up questions. We provide additional experiments and clarifications below.
> > >
> > > ### R1. Comparison with State-of-the-Art (SOTA) Models
> > >
> > > Following the reviewer’s suggestion, we evaluated recent frontier models on both benchmarks: our **dynamic CAPTCHA benchmark** and the **Halligan CAPTCHA benchmark**. Specifically, we tested *GPT-5.4* and *Gemini-3-Flash-Preview*. Please note that according to the [Gemini API Documentation](https://ai.google.dev/gemini-api/docs/computer-use), *Gemini-3-Flash-Preview* is currently the most capable Gemini model that supports **native computer-use capabilities**. We were therefore unable to evaluate the *Gemini-3-Pro* series and instead report results for *Gemini-3-Flash-Preview*.
> > >
> > > **Dynamic CAPTCHA Benchmark Results**
> > > |Model|Text|Compact Text|Icon Match|Icon Selection|Paged|Slider|Image Grid|Overall|
> > > |---|---:|---:|---:|---:|---:|---:|---:|---:|
> > > |GPT-5.4|58.94|55.63|92.25|94.85|30.82|37.76|29.29|**56.80**|
> > > |Gemini-3-Flash-Preview|62.91|71.83|89.44|81.62|31.51|79.72|10.71|**59.90**|
> > >
> > > **Halligan CAPTCHA Benchmark Results**
> > > |CAPTCHA Type|GPT-5.4|Gemini-3-Flash-Preview|
> > > |---|---:|---:|
> > > |amazon/waf|56.00|53.00|
> > > |funcaptcha/card|36.00|50.00|
> > > |funcaptcha/counting|36.00|68.00|
> > > |funcaptcha/dice_pair|36.00|56.00|
> > > |funcaptcha/galaxies|100.00|99.00|
> > > |funcaptcha/hand_number|19.00|36.00|
> > > |funcaptcha/rotated|77.00|94.00|
> > > |funcaptcha/square_icon|78.00|76.00|
> > > |arkose/3d_rollball|27.00|14.00|
> > > |arkose/dice_match|23.00|31.00|
> > > |arkose/numbermatch|34.00|28.00|
> > > |arkose/orbit_match|36.00|33.00|
> > > |arkose/rockstack|24.00|28.00|
> > > |baidu/rotate|16.00|31.00|
> > > |botdetect|50.00|68.00|
> > > |geetest/gobang|0.00|22.00|
> > > |geetest/icon|68.00|30.00|
> > > |geetest/iconcrush|1.00|33.00|
> > > |geetest/slide|38.00|41.00|
> > > |hcaptcha|70.00|67.00|
> > > |lemin|4.00|25.00|
> > > |mtcaptcha|42.00|58.00|
> > > |recaptchav2|77.00|9.00|
> > > |tencent/vtt|48.00|77.00|
> > > |yandex/kaleidoscope|7.00|9.00|
> > > |yandex/text|69.00|68.00|
> > >
> > > These results show that even frontier models fail to reliably solve CAPTCHAs, confirming that the problem remains open. More specifically:
> > >
> > > **1) Frontier GUI agents do not solve CAPTCHA reliably.**
> > > On our dynamic benchmark, *GPT-5.4* achieves **56.8%** and *Gemini-3-Flash-Preview* achieves **59.9%**, both far from perfect and below *ReCAP-8B* and *ReCAP-32B*, despite their substantially larger model size. Notably, both models drop below **35%** on *Paged* and *Image Grid* tasks, indicating that multi-step interaction remains a key bottleneck. This also suggests that the previously reported ~30% baseline is not an artifact of weak models, but reflects the intrinsic difficulty of interactive CAPTCHA solving.
> > >
> > > **2) Stronger frontier models do not remove the research motivation.**
> > > Results such as “~70” on Android Control are not directly comparable to our setting, as they involve different tasks and benefit from substantially larger models. In contrast, our results show that *ReCAP-32B* can retain general GUI performance while acquiring CAPTCHA-solving ability through targeted training.
> > >
> > > On portions of the Halligan CAPTCHA benchmark, frontier models outperform ReCAP-32B, which is expected given their substantially larger scale and training data. Our goal is not to outperform all frontier models, but to show that **a GUI agent can acquire transferable CAPTCHA-solving ability through targeted training on a small set of fundamental interaction primitives**. This provides a practical training recipe that can be incorporated into future large-scale GUI agents.
> > >
> > > **3) Our contribution focuses on native policy learning rather than a separate solver pipeline.**
> > > This distinction is important. Many prior CAPTCHA-solving systems rely on specialized solver pipelines or prompt engineering. In contrast, we investigate whether CAPTCHA-solving capabilities can be integrated directly into the policy of a GUI agent while preserving general GUI ability. This is particularly important for GUI agents, which may encounter diverse and previously unseen CAPTCHA variants where external solvers are impractical.
> > >
> > > ---
> > >
> > > ### R2. Distinguishing from the *Reasoning under Vision* paper
> > >
> > > We agree that both works involve reasoning-related traces. However, our claim is not that supervised fine-tuning on such traces is itself novel. Instead, ReCAP contributes a **dynamic interactive CAPTCHA training environment**, **closed-loop executable GUI trajectory generation**, and **retrospective self-correction supervision from failed agent rollouts**. In contrast, *Reasoning under Vision* primarily studies reasoning behavior through prompting and evaluation. Our focus is on **training-time policy learning for GUI agents**, rather than prompting-time reasoning or benchmark design.
> > >
> > > We thank the reviewer for acknowledging our explanation regarding MCA-Bench. In the revised manuscript, we will make this distinction clearer and describe our differences from prior work more precisely. We sincerely hope these clarifications help address the reviewer’s concerns.

---

### Official Review · Reviewer_xfsq · 2026-03-10

**Soundness:** 3
**Presentation:** 3
**Significance:** 4
**Originality:** 4
**Overall Recommendation:** 4
**Confidence:** 4

**Summary:**

This paper focuses on the shortcomings of GUI agents in CAPTCHA solving scenarios. With the rapid development of VLM, general-purpose GUI agents such as UI-TARS, OpenCUA, and Qwen3-VL can handle most tasks, including planning, reasoning, and grounding. However, some unusual nodes, such as errors and CAPTCHAs, still hinder the automation of the model. On the other hand, while dedicated CAPTCHA solving frameworks such as Halligan and Oedipus perform well on specific CAPTCHA types, they rely on a large number of API calls and external heuristic searches, resulting in high latency and inability to handle general GUI tasks. Against this background, this paper proposes a dynamic CAPTCHA generation system and an automated reasoning-action data generation pipeline. The system provides rich CAPTCHA examples, including Text, Compact Text, IconMatch, Icon Selection, Paged, Slider, and Image Grid. By constructing rich positive and negative samples, the system trains and adjusts the GUI agent's thinking and action processes, thereby obtaining an agent with native strong CAPTCHA solving capabilities without affecting its basic capabilities.

**Compliance With Llm Reviewing Policy:**

Affirmed.

**Final Justification:**

The author's reply has resolved my issue.

**Key Questions For Authors:**

* Please explain in detail how the ground truth actions for each CAPTCHA type are generated. What atomic action types are included in the action space? What are the parameter formats for each action? For scenarios requiring complex action sequences, what are the length and fault tolerance mechanisms for the action sequences? Will the differences in action sequence complexity among different CAPTCHA types lead to systemic differences in difficulty?
* What are the mechanisms and guarantees for maintaining the performance of the general GUI? The training data contains more than twice the number of CAPTCHA trajectories (approximately 160,000) compared to the general GUI data (approximately 73,000). Given this imbalance, why is the general performance of the 32B model unaffected while the 8B model shows a significant decline? Have ablation experiments been conducted to adjust the data mixing ratio?
* The seven CAPTCHA types defined in the paper differ significantly in dimensions such as required interaction primitives, action sequence length, and accuracy requirements. For types requiring long, complex sequences of actions with no room for error, the inherent difficulty is significantly higher than for single-step operation types. It is necessary to classify and discuss the different categories at the dataset attribute level, analyzing how the structural differences in the task definitions themselves affect model performance, rather than simply presenting numerical differences in the results.


The lack of an intuitive description of ground truth (GT) actions is a significant deficiency for a paper that proposes a dataset and data pipeline. If the authors could provide sufficient supplementation, the score could be improved accordingly.

**Limitations:**

* Significant bottlenecks remain in real-world transfer capabilities. Although this paper has taken an effective step in synthetic-to-real-world transfer, Table 2 shows that ReCAP still lags far behind Halligan on various real-world CAPTCHAs, indicating that the domain gap between synthetic data and real-world distributions remains significant, and generalization and sustainability remain core challenges.

**Strengths And Weaknesses:**

### Strengths
* The authors propose random rendering and Unbounded Generation methods to generate diverse CAPTCHA data, thereby facilitating the construction of CAPTCHA datasets. They consider not only the CoT process and actions of positive samples but also how the model reflects on and corrects erroneous actions, fully taking into account the problem scenarios and enhancing the robustness of the data pipeline and the model.
* The ReCAP model demonstrated effective transfer capabilities on real-world CAPTCHA tests. In real-world CAPTCHA evaluations, the ReCAP model significantly outperformed the baseline model on a variety of interaction-intensive tasks in a zero-shot setting, proving that skill transfer from synthetic data to real-world distributions is indeed feasible.
* The authors explicitly state that this paper focuses only on the logical solution aspect of CAPTCHA and does not simulate human behavioral characteristics such as mouse movement speed, pauses, or hesitations. However, some CAPTCHA safety systems rely precisely on these behavioral signals to detect robots. This discussion enriches the scope of the work and points the way for future research.

### Weaknesses
* As a work whose core contributions are dataset construction and data pipelines, the paper fails to provide any formal definition of the action space. Readers are left without a clear understanding of which atomic action types are supported, whether coordinates are absolute pixels or normalized relative coordinates, how the parameters of the drag action are represented, and the typical action sequence lengths and structures corresponding to different CAPTCHA types. This information is scattered throughout the appendix prompt text and lacks a systematic description. This is a significant flaw in the paper's writing.
* The data annotation used Qwen2.5-VL-72B-Instruct as the expert inference model, while the training target was the Qwen3-VL series models. Although the teacher model has a larger number of parameters, the knowledge distillation effect across model generations is not obvious. Does its generated inference trajectory exhibit systematic biases or error patterns? Why not use a stronger model from the same generation as the teacher, such as the larger MoE model from the Qwen3-VL series? The choice of teacher model directly impacts the dataset quality, but the paper lacks experimental discussion or comparison on this aspect.
* How does the training paradigm used handle general GUI tasks? Why doesn't it affect the planning, reasoning, and localization capabilities of general GUI tasks? The authors extracted a total of 73,000 general GUI data points from Aguvis and AgentNet, which is far less than the total of 160,000 CAPTCHA data points. How can the training be guaranteed not to collapse? The 8B model already performs significantly worse than the baseline on general GUI tasks, and the authors have not provided a reasonable explanation, attributing it only to model size.
* The paper repeatedly emphasizes that ReCAP is more efficient than Halligan and mentions that ReCAP's average execution time is less than 3 seconds, but it lacks systematic quantitative data. What is the end-to-end time distribution across different CAPTCHA types? What are the respective proportions of model inference time, screenshot acquisition time, and action execution time? What is the timeout failure rate on real-world CAPTCHAs with strict time limits?
* No qualitative or quantitative analysis was performed on the failed cases. In which specific scenarios is the model prone to errors?

---

> ### Author Rebuttal · Authors · 2026-03-31
>
> We thank the reviewer for the constructive feedback. We respond to the concerns below.
>
> ### R1. Formal Definition of Action Space and GT Actions
> We appreciate the reviewer’s request for a formal definition of our action space. Our agent follows the Qwen3-VL action format, extended to support multiple action outputs. The pipeline uses five main GUI actions: left click, typing, scrolling, mouse move, and left click drag. They are represented as:
> 1. Left click: `{"name":"computer_use","arguments":{"action":"left_click","coordinate":[x,y]}}`
> 2. Type: `{"name":"computer_use","arguments":{"action":"type","text":"text"}}`
> 3. Scroll: `{"name":"computer_use","arguments":{"action":"scroll","pixels":x}}`
> 4. Mouse move: `{"name":"computer_use","arguments":{"action":"mouse_move","coordinate":[x,y]}}`
> 5. Left click drag: `{"name":"computer_use","arguments":{"action":"left_click_drag","coordinate":[x,y]}}`
>
> For drag, the agent first utilize `move` to move the cursor the initial coordinates, then executes `left_click_drag` to hold the cursor and transition to the target position. Other actions are atomic.
>
> We also support multiple sequential actions in a single output by separating `computer_use` calls with commas inside `<tool_call>`. For example:
>
>     <tool_call>
>     {"name":"computer_use","arguments":{"action":"left_click","coordinate":[104,516]}},
>     {"name":"computer_use","arguments":{"action":"left_click","coordinate":[204,892]}}
>     </tool_call>
>
> We will add complete details to the appendix.
>
> ---
>
> ### R2. Interaction Complexity
> We define interaction complexity as the number of executable GUI actions required to complete a CAPTCHA. Some challenges are solvable in one or two actions, while others require longer trajectories with card switching, dragging, refinement, and confirmation. To improve robustness under longer horizons, our system includes: (1) an error-tolerant action parser that recovers from minor format inconsistencies, and (2) iterative self-correction that lets the model revise imperfect intermediate actions.
>
> ---
>
> ### R3. CAPTCHA Difficulty Analysis
> We agree that action-sequence complexity systematically affects difficulty, since longer trajectories are more prone to error accumulation. To quantify this in a controlled setting, we evaluate 500 Paged CAPTCHA instances with ReCAP-32B and group them by interaction length: direct completion (first card already correct), moderate interaction (1–2 card switches), and high interaction (>2 card switches). Direct-completion cases achieve a 2.3% higher success rate than moderate cases and a 4.1% higher success rate than high-interaction cases. These results support that longer action horizons make CAPTCHA solving harder with compounding errors in perception, grounding, and control.
>
> ---
>
> ### R4. General GUI Preservation under CAPTCHA-Heavy Training
> We clarify that our method does **not** provide a formal guarantee of preserving general GUI performance under CAPTCHA-oriented finetuning; preservation is currently supported empirically. We discussed more tradeoffs with capacity and generalization in our reply (R5) to Reviewer QmtB.
>
> We do not fix the CAPTCHA/general-GUI mixture ratio a priori. Instead, we progressively increase the CAPTCHA proportion and choose the final mixture based on an empirical saturation point:
> |CAPTCHA Data Scale|Overall Success (%) ↑|Avg Steps ↓|
> |---|---:|---:|
> |50%|62.90|1.60|
> |100%|71.90 |1.54|
> |150%|72.20 |1.52|
> Increasing CAPTCHA data improves performance up to a saturation point, beyond which gains are marginal. We guide the final mixture base on these experiments.
>
> To probe model capacity, we additionally tuned **Qwen3-VL-4B-Thinking** during rebuttal with the same setting in the paper. The 4B model shows degradation on grounding benchmarks, with the largest relative drop on ScreenSpot-V2 (−21.15% vs. baseline). Combined with our existing results on 8B models, this suggests smaller models are more sensitive to the trade-off between CAPTCHA specialization and general GUI retention, whereas larger models (32B) preserve general capabilities better under the same training regime.
>
> ---
>
> ### R5. Real-World Transfer
> We agree that real-world CAPTCHA performance is important. Our main claim is **zero-shot transfer**: ReCAP is not trained on the Halligan real-world benchmark, yet it consistently improves over its base model and remains robust across CAPTCHA types. This is especially clear on our dynamic CAPTCHA benchmark, where Halligan CAPTCHA solver’s success rate drops substantially. In contrast, ReCAP performs strongly on held-out dynamic tests and even outperforms Halligan CAPTCHA solver in several categories on its own benchmark. This suggests that ReCAP learns transferable CAPTCHA-solving ability rather than overfitting to a single dataset. We will revise the paper to clarify that the “30%→80%” improvement refers to the synthetic setting, while the real-world results should be interpreted as evidence of zero-shot generalization.

---

> > ### Author Rebuttal · Reviewer_xfsq · 2026-04-02
> >
> > The author's reply has resolved my issue. Specifically regarding the definition of CAPTCHA actions, I hope the author will add examples in the revision or appendix as promised. I will increase the score accordingly.

---

### Official Review · Reviewer_c4X2 · 2026-03-11

**Soundness:** 3
**Presentation:** 4
**Significance:** 3
**Originality:** 4
**Overall Recommendation:** 5
**Confidence:** 3

**Summary:**

The authors present ReCAP, an end-to-end framework designed to equip native Vision-Language Model (VLM) GUI agents with robust CAPTCHA-solving capabilities without degrading their general-purpose interaction skills. The paper introduces three primary contributions:
1. Dynamic CAPTCHA System: A programmatic environment that procedurally generates seven representative interactive CAPTCHA variants (e.g., Image Grid, Slider, Icon Match) utilizing stochastic rendering to prevent layout overfitting.
2. Scalable Data Curation Pipeline: An automated pipeline that generates large-scale "Reasoning-Action" (Chain-of-Thought) solution traces. Crucially, it also introduces a "Self-Correction" trace generation mechanism, which leverages failed trajectories from a student model to teach the agent error-recovery.
3. Unified Training Paradigm: A joint loss formulation that equalizes the weight of reasoning and action tokens ($\lambda_{think} = \lambda_{act} = 0.5$) during autoregressive training, preventing the model from over-indexing on linguistic generation at the expense of precise GUI actions.

**Compliance With Llm Reviewing Policy:**

Affirmed.

**Final Justification:**

The authors have resolved all my concerns. I will remian my previous recommendation.

**Key Questions For Authors:**

See Weakness.

**Limitations:**

Yes.

**Strengths And Weaknesses:**

**Strengths:**

* 1. A common pitfall in task-specific agent tuning is catastrophic forgetting of general capabilities. The authors effectively demonstrate that tuning a sufficiently large model (Qwen3-VL-32B) on structured reasoning-action CAPTCHA data preserves, and in some cases marginally improves, performance on general GUI benchmarks like Android Control and MM-Mind2Web.

* 2. The transition from synthetic training data to real-world evaluation is rigorous. ReCAP-32B demonstrates highly competitive zero-shot transfer to in-the-wild challenges from major providers (e.g., Google's reCAPTCHA v2, Arkose Labs), frequently outperforming specialized pipelines like Halligan.

* 3. By framing CAPTCHA solving as a native, multi-action generation task within a single response, ReCAP dramatically reduces the latency and interaction steps required compared to heuristic-based or search-heavy frameworks. ReCAP-32B requires only an average of 1.54 steps, which is critical for bypassing CAPTCHA time-out windows.

* 4. The inclusion of distinct ablation studies for both the CoT reasoning traces and the self-correction mechanism effectively validates the core hypotheses of the data generation pipeline.


**Weakness:**

* 1. The evaluation robustly demonstrates the model's ability to output precise target coordinates and solve the logical components of the puzzles. However, real-world CAPTCHA risk-analysis engines increasingly rely on behavioral biometrics, such as continuous mouse trajectories, acceleration, and movement hesitation. Could the authors clarify if ReCAP's generated actions can be seamlessly translated into human-like continuous control trajectories? Any preliminary demonstrations or discussions on how this behavioral gap might be bridged in future iterations would be highly valuable.

* 2. The performance degradation of the ReCAP-8B model on general GUI benchmarks highlights an interesting capacity-generalization trade-off when learning these specific interaction primitives. Could the authors elaborate on whether they have explored, or plan to explore, parameter-efficient continuous learning techniques to mitigate this? For instance, would isolating the CAPTCHA-specific tuning to a geometric trust region or utilizing specific LoRA modules help protect the base GUI representations in smaller capacity models?

---

> ### Author Rebuttal · Authors · 2026-03-30
>
> We thank the reviewer for the insightful comments. We appreciate the positive feedback on the framework's efficiency, the rigor of our real-world evaluation, and the significance of our contributions. We respond to the specific concerns regarding behavioral biometrics and capacity trade-offs below.
>
> ---
>
> ### R1. Behavioral Biometrics Translation
>
> We thank the reviewer for raising this point. In the current work, ReCAP primarily focuses on learning the perception, reasoning, and interaction primitives needed to solve modern interactive CAPTCHAs, including continuous-control tasks such as slider alignment and drag-based matching. Our current action representation is therefore best viewed as a high-level control policy that predicts *what* interaction to perform and *where*, rather than a full human-motor model that reproduces realistic low-level cursor dynamics.
>
> That said, we believe this gap between visual and behavioral CAPTCHAs is bridgeable. A natural next step is to treat ReCAP’s predicted actions as high-level intent and compile them through a downstream motion controller that generates smooth, extended trajectories with controllable velocity, acceleration, and pause profiles. We will clarify this limitation as an important direction for future work.
>
> ---
>
> ### R2.Capacity-Generalization Trade-off
>
> We appreciate this insightful observation and agree with the reviewer’s interpretation. Our results suggest a clear capacity-generalization trade-off: the 32B model largely preserves general GUI capabilities after CAPTCHA finetuning, whereas the 8B model shows consistent degradation on standard GUI benchmarks. Our current hypothesis is that, in smaller models, jointly learning CAPTCHA-specific interaction primitives and general GUI behaviors introduces representational interference. In contrast, the larger model has sufficient capacity to absorb the new capability with much less disruption to the base GUI representations.
>
> In the current version, we partially mitigate this effect by mixing general GUI trajectories into training, but we have not yet performed a systematic study of parameter-efficient continual learning methods. We agree this is a promising direction. In particular, isolating CAPTCHA-specific adaptation into dedicated LoRA/adaptor modules, constraining updates to a trust region around the base GUI policy. We believe it's an important direction for future work.
>
> ---
>
> We hope these clarifications resolve your questions, and we will greatly appreciate your continued support of our paper. Please feel free to reply if you require further discussion on any of these points. We are more than happy to provide additional detail.

---

> > ### Author Rebuttal · Reviewer_c4X2 · 2026-04-02
> >
> > Thanks for the new clarifications. The rebuttals have resolved all my concerns. I will remian my previous recommendation.

---

### Official Review · Reviewer_QmtB · 2026-03-11

**Soundness:** 2
**Presentation:** 3
**Significance:** 3
**Originality:** 3
**Overall Recommendation:** 4
**Confidence:** 3

**Summary:**

This paper presents ReCAP, a system for training native GUI agents to solve modern interactive CAPTCHAs. The authors build a dynamic CAPTCHA generation system covering seven challenge types, develop an automated pipeline for generating reasoning-action trajectories and self-correction traces, and finetune Qwen3-VL models with a weighted loss balancing reasoning and action tokens. ReCAP-32B improves CAPTCHA success from ~30% to ~80% on held-out tests while maintaining general GUI performance.

**Compliance With Llm Reviewing Policy:**

Affirmed.

**Final Justification:**

The authors' reply has basically addressed my concerns.

**Key Questions For Authors:**

Please refer to the weaknesses mentioned above. The reviewer is willing to raise the score if they are addressed satisfactorily.

**Limitations:**

Yes.

**Strengths And Weaknesses:**

# Strengths

S1: **Well-motivated and practical problem.** The paper identifies a genuine gap: general GUI agents fail at CAPTCHAs, and specialized CAPTCHA solvers can't do general GUI tasks. The framing of CAPTCHA solving as a set of learnable interaction primitives that complement general GUI skills is clean and well-articulated.

S2: **Comprehensive dynamic CAPTCHA system.** The seven CAPTCHA types are well-chosen to cover distinct interaction modalities. The stochastic rendering is a sensible design choice to prevent overfitting. The meta API providing ground truth enables scalable automated data generation.

S3: **Self-correction trace generation is a compelling idea.** Using failed student attempts to generate corrective reasoning traces is well-motivated for CAPTCHA solving, where multi-step interaction and error recovery are inherent.


# Weaknesses
W1: **Evaluation on the authors' own synthetic benchmark is the primary result, raising circularity concerns.** The strongest results (Table 1) are on the dynamic CAPTCHA system that the authors designed and used for training data generation. While the held-out test set uses disjoint instances, the visual distributions, interaction patterns, and challenge structures are identical to those of training. The real-world evaluation (Table 2) is more informative but shows much more modest and mixed results, as on many CAPTCHA types, absolute success rates remain low. The paper's headline claim of "30% → 80%" is thus primarily about the synthetic setting and somewhat overstates the practical advance.

W2: **General GUI evaluation is limited to easier benchmarks, while more challenging ones are omitted.** The paper evaluates general GUI capabilities on Android Control, ScreenSpot-V2, and MM-Mind2Web, which are relatively standard and moderate-difficulty benchmarks. Notably absent are more challenging and recent GUI benchmarks such as ScreenSpot-Pro[1] and UI-Vision[2], which test fine-grained grounding on professional software interfaces (e.g., CAD tools, IDEs, spreadsheets) with smaller UI elements and more complex layouts. These harder benchmarks would provide a much more rigorous test of whether CAPTCHA finetuning preserves the model's visual grounding under difficult conditions. Given that the authors claim ReCAP-32B "largely preserves general GUI agent performance," this claim would be substantially more convincing if validated on benchmarks where strong base models actually struggle. The near-ceiling performance of the Qwen3-VL-32B baseline on ScreenSpot-V2 (94.5%) leaves little room to detect degradation, while a harder benchmark would be more sensitive to any loss in fine-grained visual perception caused by the CAPTCHA-focused finetuning.

[1] Li et al. ScreenSpot-Pro: GUI Grounding for Professional High-Resolution Computer Use.

[2] Nayak et al. UI-Vision: A Desktop-centric GUI Benchmark for Visual Perception and Interaction.

W3: **The reasoning trace generation pipeline has a subtle issue.** The expert model is given ground truth and instructed to "reason as if the ground truth were unknown" while being constrained to produce the correct answer. This is essentially asking the model to rationalize a known answer, not genuinely reason. While automatic checks verify that the reasoning doesn't directly copy the answer, the resulting traces may contain post-hoc justifications rather than authentic reasoning chains. The paper doesn't analyze the quality or faithfulness of these generated reasoning traces.

W4: **Limited analysis of failure cases and generalization boundaries.** The paper reports aggregate success rates but provides little insight into why the model fails on the remaining ~20% of synthetic cases or the many real-world CAPTCHA types where performance is low. Understanding failure modes would be valuable for assessing the approach's limitations and guiding future work.

W5: **The capacity-generalization trade-off for ReCAP-8B is concerning but underexplored.** Table 3 shows consistent degradation on general GUI benchmarks for ReCAP-8B (e.g., ScreenSpot-V2 drops from 92.06% to 80.03%). This suggests that the approach may not scale down gracefully, which limits its practical applicability.

---

> ### Author Rebuttal · Authors · 2026-03-30
>
> We thank the reviewer for the constructive feedback. We appreciate the positive comments on the motivation, CAPTCHA design, and data pipeline. We respond to the main concerns below.
>
> ---
>
> ### R1. Synthetic vs. real-world evaluation
>
> We agree that real-world CAPTCHA is an important setting. Our main claim is **zero-shot transfer**: ReCAP is not trained on the Halligan real-world benchmark, yet it consistently improves over its base model and performs strongly across many CAPTCHA challenges. This is especially clear when compared with Halligan on our dynamic CAPTCHA benchmark, where Halligan’s performance drops substantially. In contrast, ReCAP performs strongly on our held-out dynamic benchmark and also outperforms Halligan on multiple CAPTCHA challenges in Halligan’s own benchmark. We believe this supports our claim that ReCAP learns transferable CAPTCHA-solving skills rather than overfitting to one benchmark.
>
> We also agree there is room for improvement. Better interactive primitive design and richer training diversity should further improve performance on complex CAPTCHA variants never seen during training. We will revise the “30%→80%” phrasing to make clear that it refers to the synthetic setting, while framing the real-world results as evidence of zero-shot transfer.
>
> ---
>
> ### R2. General GUI evaluation
>
> We acknowledge the concern regarding the limitations of general GUI agent benchmarks. To address this, we conducted additional evaluation with UI-Vision on the ReCAP model and the baselines. Although UI-Vision does not officially support Qwen3-style bounding box outputs, we developed a custom parser to facilitate this evaluation. We run all models with thinking disabled and show the results below:
>
> **Element Evaluation**
> |Model|Basic Acc|Functional Acc|Spatial Acc|Overall Acc|
> |---|---:|---:|---:|---:|
> |Qwen3-VL-Thinking-32B|16.93%|15.80%|7.39%|13.20%|
> |ReCAP-32B|16.59%|15.74%|6.67%|12.81%|
> |Qwen3-VL-Thinking-8B|9.76%|9.31%|3.72%|7.48%|
> |ReCAP-8B|10.67%|8.86%|4.13%|7.78%|
>
> **Layout Evaluation**
> |Model|Mean IoU|Mean Precision|Mean Recall|
> |---|---:|---:|---:|
> |Qwen3-VL-Thinking-32B|0.488|0.661|0.620|
> |ReCAP-32B|0.498|0.657|0.683|
> |Qwen3-VL-Thinking-8B| 0.331|0.708|0.410|
> |ReCAP-8B|0.305|0.442|0.565|
>
> The results show that ReCAP remains broadly comparable to the base models on a harder grounding benchmark. At 32B, element accuracy is nearly unchanged, and layout IoU is slightly improved with higher recall. At 8B, we observe trade-offs across metrics, but not a collapse of grounding ability. This further supports that ReCAP largely preserves general GUI capability, especially at 32B scale.
>
> ---
>
> ### R3. Reasoning trace generation
> We agree that directly generating reasoning without annotation would be cleaner. Our goal, however, is to provide **structured intermediate supervision** that better aligns perception, reasoning, and action. Similar to [STaR](https://arxiv.org/abs/2203.14465) (Zelikman et al., 2022), which conditions rationale generation on the correct answer when unconstrained rationale generation is unreliable, we prioritize **usefulness and scalability** over reasoning style quality during large-scale data generation. Manual inspection of 100 sampled traces found them generally coherent and task-aligned. We will revise the paper to clarify this trade-off.
>
> ---
>
> ### R4. Failure analysis
> We agree that a deeper analysis would strengthen the paper. We manually inspected failed samples from the held-out test set. The most frequent error modes are:
> 1. **Highly distorted OCR**: severe distortion remains hard even after several correction attempts; common confusions include “1” vs. “I” and “0” vs. “O”.
> 2. **Multi-step error accumulation**: an early mistake causes later corrections to drift further off track.
> 3. **Image recognition errors**: the model misidentifies icons or visual elements.
> These failures suggest that the main bottlenecks are **perception robustness** and **long-horizon recovery**. We plan to strengthen the corresponding interactive primitives in future work and we believe it can further improve ReCAP’s performance. We will also add the failure case examples to the appendix.
>
> ---
>
> ### R5. Capacity-generalization trade-off
> ReCAP-8B shows that capability integration is not free: smaller models are more vulnerable to interference between CAPTCHA-specific skills and general GUI behaviors. In contrast, ReCAP-32B largely preserves general GUI performance while gaining substantial CAPTCHA ability. We will clarify that 32B is the primary evidence for successful capability integration, while 8B is a lower-capacity stress test.
>
> ---
>
> Overall, we believe the results support the main message of the paper: CAPTCHA solving can be internalized as a transferable skill in native GUI agents, with strong zero-shot transfer and preserved general GUI ability at sufficient model scale. **We hope our response addresses your concerns. We’re happy to discuss in more detail should you have more questions.**

---

> > ### Author Rebuttal · Reviewer_QmtB · 2026-04-03
> >
> > Thank the authors for the detailed reply, which has basically addressed my concerns. Though there are still some minor limitations in the scale of the work, I feel it is a valid contribution to the community, so I have raised my score to 4. Please include the content in the rebuttal (especially R1, R2, and R4) in the later version to make the work more rigorous.

---

### Decision · Program_Chairs · 2026-04-30

**Decision:**

Accept (regular)

**Comment:**

This submission proposes ReCAP, a native GUI agent that learns CAPTCHA solving through a dynamic interactive CAPTCHA environment, automatically generated reasoning-action trajectories, and retrospective self-correction traces from failed rollouts. Across the reviews, there was broad agreement that the systems/data contribution is substantial. The strongest positive points were the breadth of the dynamic environment, the scalable closed-loop data generation pipeline, the explicit self-correction mechanism, and the ablations showing that both reasoning traces and correction traces contribute to the reported gains. The main reservations were that the headline “30%→80%” claim is primarily supported by the authors’ own held-out synthetic benchmark; real-world transfer is more mixed; the original evidence for preserving general GUI skill was limited, especially for harder benchmarks and for smaller models; the reasoning traces are answer-conditioned and may therefore be partially post hoc; and the original manuscript under-positioned itself relative to prior CAPTCHA benchmarks and did not sufficiently clarify action space and training details.

After rebuttal, three reviewers stated that their concerns were fully resolved. The authors added harder GUI evaluation on UI-Vision, failure-mode analysis, a formal action-space description, additional data-mixing/capacity evidence, and frontier-model comparisons on both the dynamic benchmark and the Halligan benchmark. These additions substantially strengthen the paper. That said, several limitations remain and should be explicitly reflected in the final version: the strongest claim should be narrowed to the held-out synthetic setting; real-world generalization remains mixed and is not uniformly state of the art once frontier baselines are included; the faithfulness of the answer-conditioned reasoning traces remains unverified; and some evaluation gaps remain, including no Oedipus comparison, no full latency decomposition, and no real ablation of the reasoning/action loss weights.

Reviewer XN4J raised the most substantial objections: inaccurate positioning with respect to MCA-Bench/Open CaptchaWorld, insufficient support for the loss-weight choice and missing training details, lack of state-of-the-art closed-model baselines, limited originality relative to concurrent reasoning-based CAPTCHA work, and overstatement of the size of the research gap. The rebuttal and follow-up addressed several of these points: the authors acknowledged the positioning issue, promised clearer manuscript revisions, added training details, and supplied GPT-5.4 and Gemini-3-Flash-Preview results. However, this reviewer remained only partially satisfied, arguing that the added baselines materially shrink the apparent gap and weaken the novelty claim. The reviewer stated that they would raise the score, but still could not support the paper. AC agrees with Reviewer XN4J's concern that "the central conclusion and the claimed contribution of the paper are materially changed as a result." Therefore, the 30->80 gain is a misleading overclaim. Nonetheless, over-focusing on comparison with proprietary models does to another extreme, and under estimates the paper's contribution in providing a working recipe for the problem.

I recommend Weak Accept because, despite meaningful remaining limitations, the paper makes a credible and useful contribution by showing that CAPTCHA-solving skill can be internalized within a native open GUI-agent policy through a dynamic training environment, automatically generated reasoning-action/self-correction data, and multi-step action training. Author should make major changes to tune down the claims and revise the paper according to the rebuttal discussions.